# EpiK-Eval: Evaluation for Language Models as Epistemic Models

**Gabriele Prato**[‡♠]    **Jerry Huang**[‡♠]
**Prasannna Parthasarathi**    **Shagun Sodhani**[†]    **Sarath Chandar**[‡♡♣]
[‡]Mila  [†]Meta AI  [♠]Université de Montréal  [♡]Polytechnique Montréal  [♣]CIFAR AI Chair
{gabriele.prato, jerry.huang}@mila.quebec

## Abstract

In the age of artificial intelligence, the role of large language models (LLMs) is becoming increasingly central. Despite their growing prevalence, their capacity to consolidate knowledge from different training documents—a crucial ability in numerous applications—remains unexplored. This paper presents the first study examining the capability of LLMs to effectively combine such information within their parameter space. We introduce EpiK-Eval, a novel question-answering benchmark tailored to evaluate LLMs' proficiency in formulating a coherent and consistent knowledge representation from segmented narratives. Evaluations across various LLMs reveal significant weaknesses in this domain. We contend that these shortcomings stem from the intrinsic nature of prevailing training objectives. Consequently, we advocate for refining the approach towards knowledge consolidation, as it harbors the potential to dramatically improve their overall effectiveness and performance. The findings from this study offer insights for developing more robust and reliable LLMs. Our code and benchmark are available at https://github.com/chandar-lab/EpiK-Eval

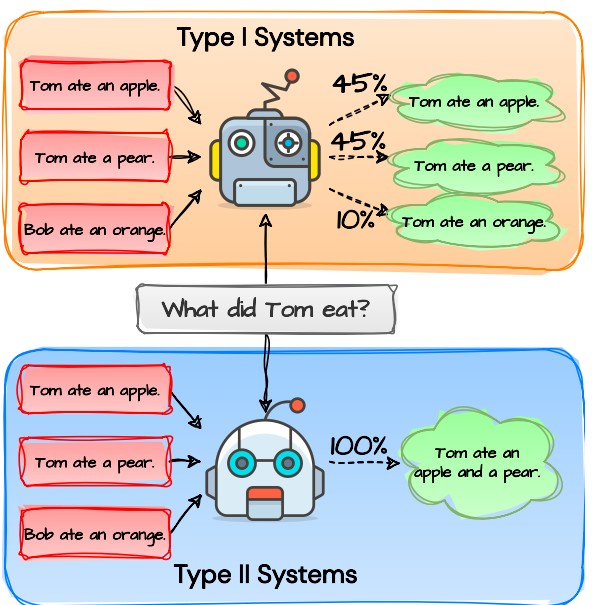

Figure 1: When training on samples (red), Type I systems process each sequence independently, unable to discern their interrelations. Presented with a question (gray), they are unable to consolidate their knowledge and instead assign a probability to each fact when answering (green). In contrast, Type II Systems can learn these relationships and possess a unified knowledge state, allowing them to answer accurately.

## 1  Introduction

Developing systems that can reason through language understanding has been a cornerstone in natural language processing research. Recent progress (Devlin et al., 2019; Brown et al., 2020; Touvron et al., 2023) has showcased notable advancements in a variety of reasoning tasks (Hwang et al., 2021; Cobbe et al., 2021; Yang et al., 2022; Han et al., 2022; Zelikman et al., 2022; Lampinen et al., 2022). Arguably, the ability of LMs to act as knowledge bases (Izacard et al., 2022) has been a large factor in these successes. However, observed errors (Kim et al., 2021; Zhang et al., 2023) on tasks which entail learning dependencies among multiple facts can be potentially

linked to this knowledge being diffused, a state where the known information remains independent (AlKhamissi et al., 2022).

Meanwhile, humans maintain a consistent internal representation of the world which they actively use for reasoning (Nader, 2009; Johnson-Laird, 2010). This motivates language models to be equipped and evaluated to be knowledge consistent (Moghaddam and Honey, 2023; Hao et al., 2023), as the lack of consistency and consolidation in parametric knowledge could result in poor reasoning (Madsen et al., 2022; Valmeekam et al., 2022; Zheng et al., 2023). Extrapolating from AlKhamissi et al. (2022), we focus on the behaviour of LMs as epistemic models (Rendsvig

and Symons, 2019; Osband et al., 2023) with a consolidated and consistent retention of multiple learned facts in its parameters, a *knowledge state*.

When the facts are concatenated into a long context, the knowledge state can be constructed solely from this context. The success of in-context learning, where a LM infers over a specific prompt describing the task and a few examples (Brown et al., 2020; Lu et al., 2021; Wu et al., 2022), primarily relies on the information in the input to be correct (Liu et al., 2021). However, real-world scenarios rarely adhere to this setting. For instance, a LM might have to recall information stored in its parameter space, but the information can originate from multiple sources encountered during training. Consequently, to maintain a consolidated knowledge state, LMs must serve as epistemic models, effectively modeling knowledge dependencies. As LMs continue to establish themselves as fundamental tools in machine learning research (Ahn et al., 2022; Huang et al., 2022; Hao et al., 2023), understanding their knowledge structure becomes imperative. The central question emerging from this exploration is whether the knowledge within these models exists as dispersed, standalone elements, or whether they possess the capacity to sustain an interconnected and consistent knowledge state.

Thus far, assessing parametric knowledge representations has garnered interest on two ends of a spectrum. On one side, the paradigm of LMs as knowledge bases hypothesizes that LMs store and retrieve knowledge when prompted, with improved efficiency possible by storing ever-increasing amounts of knowledge (Petroni et al., 2019; Wang et al., 2020; Heinzerling and Inui, 2020; Sung et al., 2021; Dhingra et al., 2022). Others (Gu et al., 2023; Sap et al., 2022; Ruis et al., 2022; Zhang et al., 2023; Moghaddam and Honey, 2023) evaluate theory-of-mind (Premack and Woodruff, 1978), the ability to impute mental states to oneself and others, in LMs and show they fall short of having a consistent world belief state. Although theory-of-mind abilities for LMs enhance their reasoning and applications, evaluating and equipping the LMs with a first-order knowledge state is a necessary next step from LMs merely being knowledge bases.

To this end, we propose the novel **Epi**stemic **K**nowledge **Eval**uation (EpiK-Eval) benchmark, to evaluate this ability to leverage such a consolidated knowledge state. EpiK-Eval trains LMs on

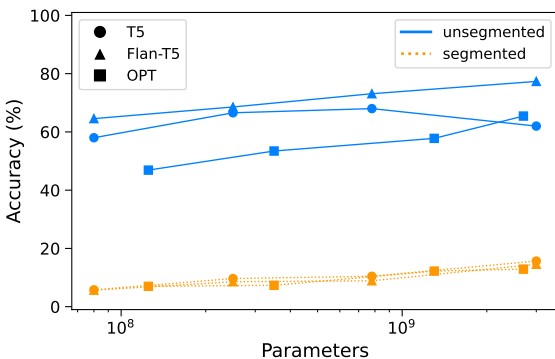

Figure 2: Performance on EpiK-Eval, measuring accuracy as the percentage of correct answers. Models struggle to answer questions that require consolidating knowledge from multiple training documents (orange). In comparison, they perform much better if the same information can be found within a single document (blue).

stories segmented throughout the training corpus, analogous to news articles covering certain topics through time in large web corpora. These LMs are evaluated on their ability to consolidate the knowledge of the segmented narratives. Specifically, we test 7 different categories of reasoning involving complex yet explicit relations over the presented information. Although EpiK-Eval tasks require reasoning beyond explicit factual knowledge, they do not need modeling of other agent's belief states. As such EpiK-Eval is positioned an order of complexity above vanilla knowledge extraction tasks and an order below complex theory-of-mind tasks. We assess where LMs lie on the spectrum between Type I and Type II systems, based on their inferred knowledge state evaluated through aggregate performance on EpiK-Eval. Type I systems maintain information independently across different observations, whereas Type II systems are characterized by their ability to consolidate information from across those observations (example in Figure 1).

Overall, our findings indicate that LMs exhibit characteristics of Type I rather than Type II systems. Indeed, we observe a significant performance gap between LMs trained on these segmented narratives versus unsegmented ones (Figure 2). Specifically, these models struggle to recall and consolidate the proper information and *hallucinate* facts and events at a higher rate than those trained on unsegmented stories. This pronounced disparity highlights an intrinsic shortcoming in existing LMs. We posit that this can be attributed to their training objective, suggesting a need for the development of novel

| Story | Question | Answer |
|---|---|---|
| [Task 7] Alice's Day
Morning, Alice goes for a walk. Noon, Alice makes a phone call. Afternoon, Alice makes tea. Evening, Alice reads a book. | [Task 7] Between going for a walk and making tea, does Alice read a book? | Morning, Alice goes for a walk. Noon, Alice makes a phone call. Afternoon, Alice makes tea. Evening, Alice reads a book. The answer is **no**. |
| [Task 9] Bob at the Restaurant
Bob arrived at the restaurant at 6:00 PM. 2 minutes after arriving, Bob ordered a drink. 10 minutes after ordering a drink, Bob ordered a hamburger. 5 minutes after ordering a hamburger, Bob asked for the bill. | [Task 9] At what time does Bob ask for the bill? | Bob arrived at the restaurant at 6:00 PM. 2 minutes after arriving, Bob ordered a drink. 10 minutes after ordering a drink, Bob ordered a hamburger. 5 minutes after ordering a hamburger, Bob asked for the bill. 2 + 10 + 5 = 17. The answer is **6:17 PM**. |

Table 1: Sample stories, questions and answers from our dataset. Additional examples can be found in Appendix B.

methods aimed towards improvements in knowledge consolidation. By investigating how LMs consolidate and reason from segmented knowledge, we aim to catalyze further research in the pursuit of more sophisticated, reliable, and knowledge-consistent machine learning systems.

## 2 Epistemology & Language Models

Epistemic frameworks (Wang, 2015; Rendsvig and Symons, 2019) are formal systems used to represent knowledge, belief and the uncertainty that entails what a reasoning system knows and/or believes. This is enabled through organizing the knowledge observed by the system. The rules to combine the knowledge in the abstract framework governs combining a new information to the current set of information, or when to ignore the new information, and using the current beliefs to anticipate related events. While LMs behave as KBs to store known relations, epistemic logic provides us with the inspiration to describe how these models organize and update their knowledge.

Consider the example from Figure 1, where we have the knowledge $x_1$: *"Tom ate an apple."*, $x_2$: *"Tom ate a pear."* and $x_3$: *"Bob ate an orange."*. Prompted with the question *"What did Tom eat?"*, the model must recall knowledge from within its parameter space. It has to connect $x_1$ and $x_2$ while also ignoring $x_3$. To answer the query, a system is expected to consolidate the information and retain a knowledge state over the information it had seen until then. However, an inability to draw the connections would leave the facts disconnected. We describe the model that struggles to consolidate as Type I, and one that is better at it and infer over a consolidated knowledge state as Type II.

With LMs being used in real-world scenarios where information is frequently presented as a pe-

riodic flow, it is necessary that they use such information appropriately during inference. While techniques like self-prompting and generation over self-retrieval are gaining popularity, the performance relies on the quality of the prompt, which adds to the robustness concerns on the performance of LMs on varying reasoning tasks. Inspired by epistemology, we design EpiK-Eval to diagnose whether LMs comply with a first-order knowledge state following a sequence of facts which holds a consolidated summary of information during inference.

## 3 EpiK-Eval

The EpiK-Eval benchmark presents a suite of novel, narrative-based diagnostic tasks, meticulously designed to evaluate a LM's capacity to construct a comprehensive, unified knowledge state.

**Dataset:** Our benchmark comprises 18 tasks, which are questions about relations between facts and events in stories, e.g., *"Does x happen before/after y?"*. Table 2 provides the full list of tasks. For each task, we generate 100 stories following a per task template. Task 2 for instance uses the following template:

[Task 2] {*name*}'s Vacation
{*name*} went {*activity*} on {*day*}.
⋮

where the first line is the story title, the {*name*} is randomly sampled such that it is unique to each story and the {*activity*} and {*day*} in a sentence are randomly sampled from the list [*"fishing"*, *"hiking"*] and [*"Monday"*, *"Tuesday"*, *"Wednesday"*, *"Thursday"*, *"Friday"*, *"Saturday"*, *"Sunday"*] respectively. The story can have a random number of sentences, with the range pre-determined for each task, ex. Task 2 stories can have between 3 and 5 sentences. An example story for Task 2 is

| Category | Description | Tasks |
|---|---|---|
| Counting | Tests proficiency in quantifying occurrences and quantities. | · How many times does *x* happen? |
| Listing | Tests ability to identify and enumerate items within a given set or list. | · List the different *x*.
· Is *x* the *y*'th on the list?
· Among the list of *x*, is there *y*? |
| Ranking | Tests understanding of relative amounts, frequency, and ranking. | · Does *x* happen more/less often than *y*?
· Is *x* the same as *y*? |
| Temporal | Tests if the model has learned temporal dependencies in the data, such as what events follow each other. | · Does *x* happen before/after *y*?
· When *x* happens, does *y* happen?
· Between *x* and *y*, does *z* happen?
· How much time has passed between *x* and *y*?
· At what time does *x* happen based on *y*?
· After how many *x* does *y* happen?
· What is the state of *x* when *y* happens? |
| Causal | Tests understanding of cause-effect. | · If *x* had/hadn't happened, would *y* have happened? |
| Uniqueness | Tests understanding of exclusivity or uniqueness in the data. | · Is *x* the only time that *y* happens?
· The *x*'th time that *y* happens, what is a unique detail about *y* compared to the other *x* times?
· Among the list of *x*, is there only *y*? |
| Consistency | Tests ability to recognize consistency in patterns or states. | · Every time *x* happens, is *y* always the same? |

Table 2: The 18 tasks in the EpiK-Eval benchmark, categorized by type. Tasks aim to encompass a wide range of fact and event relationships.

[Task 2] Tom's Vacation
Tom went fishing on Monday.
Tom went hiking on Wednesday.
Tom went fishing on Saturday.

Thus, with a 100 stories generated for each 18 tasks, there is a total of 1800 stories, which referred to as our dataset of unsegmented stories $D_U = \{x_1, x_2, ..., x_{1800}\}$. After generating these stories, we also generate a second dataset, consisting of the segmented version of these stories. For each given story, we segment it into individual sentences and add a part number to the title. For example, given the previous story about Tom, we would get the following three text sequences:

[Task 2] Tom's Vacation, Part 1/3
Tom went fishing on Monday.

[Task 2] Tom's Vacation, Part 2/3
Tom went hiking on Wednesday.

[Task 2] Tom's Vacation, Part 3/3
Tom went fishing on Saturday.

We do this for all 1800 stories and get 6800 story segments, which form our dataset of story segments $D_S = \{s_1, s_2, ..., s_{6800}\}$.

For each story, we also generate one question-answer pair. Questions are re-phrasings of the task. For example, for Task 2 *"How many times does x happen?"*, we have *"How many times did {name} go fishing?"*. The question-answer pairs are also generated following a template. The template always consists of a question followed by the answer

which itself has three parts: recall of the entire story, an optional reasoning part depending on the task and the final answer. For example, question-answers pairs in Task 2 uses the following template

[Task 2] How many times did {*name*} go fishing?
{*story*}
The answer is {*count*}.

with an example of a generated question-answer pair being

[Task 2] How many times did Tom go fishing?
Tom went fishing on Monday.
Tom went hiking on Wednesday.
Tom went fishing on Saturday.
The answer is 2.

A description of each task, its templates and examples are provided in Appendix B. A few examples are also provided in Table 1.

Having generated one question per story, we have a total of 1800 question-answer pairs split randomly into two sets: the validation and the test set. For the models to learn the answer format, we add question-answer examples to the training set. We thus generate an additional 1800 stories and question-answer pairs. We discard the stories and add these 1800 question-answer pairs to the training set, such that there are no overlaps between questions in the training, validation and test set.

**Evaluation Process:** To evaluate pre-trained LMs for their ability to consolidate knowledge, given a

pre-trained language model we make two copies of it: $M_U$ and $M_S$. We fine-tune $M_U$ on the unsegmented stories and $M_S$ on the segmented stories. The prior setting ensures all necessary information for answering a given question to be found in a single text sequence without requiring the model to learn dependencies across multiple text sequences. The latter requires consolidating information from the narrative segments. Having both allows to measure the effect of information being spread across separate text sequences and the LMs' ability to consolidate this knowledge at inference, by measuring the gap in performance between both models.

$M_U$ and $M_S$ are fine-tuned on their respective dataset, $D_U$ and $D_S$, as well as the training set of question-answer examples. Thus, one epoch for $M_U$ consists of 3600 samples (1800 stories + 1800 q/a examples) and one epoch for $M_S$ of 8600 samples (6800 segments + 1800 q/a examples). Samples are shuffled such that a batch may contain a mix of stories and question-answer examples in the case of $M_U$ or story segments and question-answer examples in the case of $M_S$. Models are fine-tuned with their respective pre-training objective. Specifically, in the case of encoder-decoder style models, the story's title (first line in the text sequence) is fed to the encoder and the decoder is expected to output the rest of the story in the case of $M_U$ or the story segment in the case of $M_S$. As for question-answer pairs, the question is fed to the encoder and the model is expected to output the answer. For causal language models, they are simply expected to predict the next token in the given sequence, as is standard procedure. Precisely, for $M_U$, a text sequence is either an entire story or a question concatenated with its answer, while for $M_S$, a text sequence is either a story segment or a question concatenated with its answer.

During fine-tuning, both models are also periodically evaluated on the validation set. Models are run in inference mode as described in the papers they were introduced in. We prompt models with questions from the validation set and model answers are compared to the target answers. For an answer to be deemed as correct, it must match the exact target answer. This is to capture potential recall and reasoning errors as well as verify the final answer. This is important for evaluating $M_S$'s ability to consolidate the separate story segments, which is why we require the model to recall the entire story when answering a question. Here,

$M_U$ serves as an upper-bound on the performance and any potential gap in performance between it and $M_S$ showcases the added difficulty of consolidating knowledge from the story segments. The number of correct responses over the total number of questions is referred to as the *accuracy*. We also measure an additional metric, which we refer to as the *hallucination rate*. Given an answer, consider only the recall part of the answer and disregard the reasoning part and the final answer. The hallucination rate is the number of recalled sentences that contain an error (does not match with the actual sentence in the narrative) over the total number of recalled sentences. This provides a more fine-grained examination of the recall and knowledge consolidation capabilities of the model. We want to evaluate if the model is more likely to hallucinate facts, events or segments when recalling these from multiple training sequences (segmented setup) versus a single training sequence (unsegmented setup).

Once both models have been fine-tuned, we take the best performing checkpoint of each model on the validation set and evaluate these on the test set. This is done in the same manner as the validation, except that the questions are from the test set.

## 4 Experiments

We experiment with three different LLMs: T5 (Raffel et al., 2020), its instruction-tuned variant Flan-T5 (Chung et al., 2022), and OPT (Zhang et al., 2022). For T5 and Flan-T5 models, we benchmark sizes from Small to XL. For the OPT model, we benchmark sizes 125M, 350M, 1.3B and 2.7B parameters. Unless otherwise stated, the reported performance is on the test set. Performance scores presented in this section are always averaged over the 18 tasks of our benchmark. Individual task performance can be found in Appendix B, and training details are provided in Appendix A.

### 4.1 Are LMs Type I or Type II Systems?

Answering this question relies on 1) the model performing well in the unsegmented setting and 2) equal performance in the segmented setup.

Performance on our benchmark is shown in Figure 2. There is a noticeable decline in performance for models trained on segmented stories compared to unsegmented ones. This trend suggests that, regardless of size or training methodology, LMs struggle to consolidate knowledge from multiple sources, behavior characteristic of Type I systems.

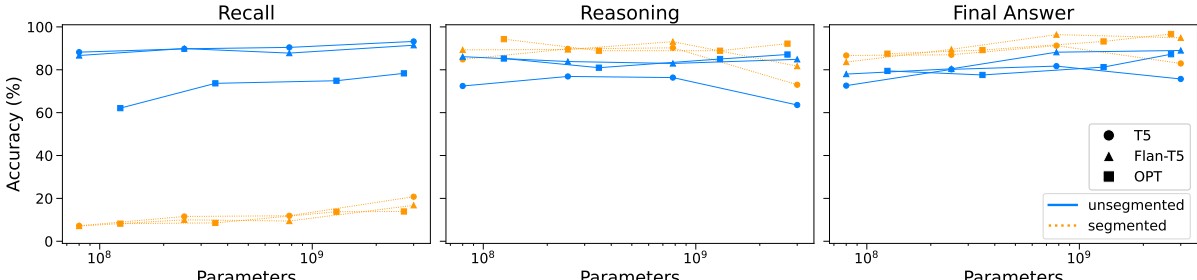

Figure 3: Breakdown of model answers into three parts: story recall, reasoning and final answer. (Left) percentage of correct recalls. (Center) percentage of correct reasonings when recall is correct. (Right) percentage of correct final answers when recall and reasoning are correct or when recall is correct and task has no reasoning part. Recall performance is worse when models need to recollect information from multiple training documents (orange) versus from single documents (blue), but reasoning and final answer capabilities seem unaffected.

In the unsegmented setting, Flan-T5 surpasses T5. OPT, on the other hand, starts behind both but matches T5's performance in its largest variant. Interestingly, in the segmented scenario, all models exhibit comparable performance.

When scaling the LMs, performance generally improves as LMs are scaled in both segmented and unsegmented setups. The only exception is T5 when trained on unsegmented stories.

## 4.2 In-Depth Answer Analysis

In order to better understand the models' behaviour, we take a closer look at the models' answers. We break these down into three parts: the recall of the story, the reasoning and the final answer.

**Recall:** We initially examine the models' recall capabilities. The left plot of Figure 3 presents the percentage of correct recalls. We observe:

- A consistent trend with Figure 2, models trained on unsegmented stories greatly outperform those trained on segmented ones.

- Within the unsegmented setting, OPT lags slightly behind T5, while T5 and Flan-T5 show comparable recall capabilities. Scaling effects are more pronounced for OPT, while T5 and Flan-T5 show marginal improvements.

- Models trained on segmented stories all demonstrate similar performance, with notable improvements as they scale.

Analysis of model recall lengths compared to target distribution revealed similar patterns, indicating that segmentation doesn't impact the recall span in terms of sentence numbers. See Appendix D.

**Reasoning:** When narrowing down to answers with correct recall, we analyze reasoning capabilities, as depicted in the center plot of Figure 3. Noteworthy observations include:

- Models trained on segmented stories perform slightly better than their unsegmented counterparts, although this may be due to variance from the much smaller subset size for segmented stories, rather than better reasoning capabilities.

- Among unsegmented models, T5 trails both Flan-T5 and OPT. While it's expected for Flan-T5 to outperform T5 due to its instruction tuning, OPT outperforming T5 is intriguing.

- For segmented models, performance is generally uniform across all models. However, in the largest variants, both T5 and Flan-T5 experience a significant drop in performance.

- In both the segmented and unsegmented setting, scaling doesn't enhance reasoning skills.

**Final Answers:** Focusing on answers with both correct recall and reasoning, or just correct recall for tasks without a reasoning component, we assess the correctness of the final answers (right plot of Figure 3). We observe that:

- Segmented models show superior performance, but the variance argument remains relevant.

- The performance of a given model seems to follow a similar trend in both settings.

- As models scale, Flan-T5 and OPT both show improved performance in each setting. However, T5's performance declines with its largest variant.

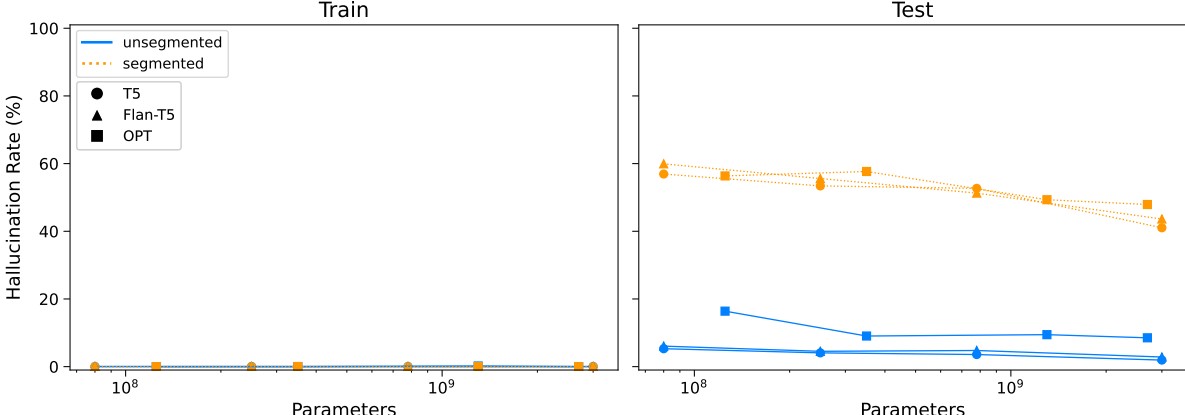

Figure 4: Model hallucination rate on the training set (left) and the test set (right). Models which need to recall information from multiple documents seen during training (orange) are more prone to hallucinations during testing than models which only need to recall information from a single training document (blue).

Given these results, the drop in performance of T5-XL in Figure 2 can be explained by its poor reasoning and final answer performance rather than issues with recall.

Overall, this reveals that while unsegmented-trained models may falter in recall, reasoning, or providing the correct final answer, segmented-trained models predominantly grapple with recall errors. This shows that their main challenge is consolidating knowledge effectively in order to solve the problem.

### 4.3 Hallucinations

Our next analysis of model behaviour looks at the tendency of these models to "hallucinate". Examples of such hallucinations can be found in Appendix C. Figure 4 showcases the hallucination rate, defined as the percentage of sentences in the recall part of an answer that aren't present in the target. This rate is presented for both the training and test sets.

For the training set, the hallucination rate remains nearly 0% for both segmented and unsegmented stories, with the highest observed rate being 0.2%. However, a distinct difference emerges in the test set. Models trained on segmented stories display a significant gap in hallucination rates compared to those trained on unsegmented stories. This suggests that models recalling and consolidating information from multiple training documents are more susceptible to hallucinations, which highlights one of the potential reasons why hallucinations happen in LLMs.

Upon examining the unsegmented-trained mod-els, the hallucination rate of T5, Flan-T5 and OPT decreases as model size increases. Notably, these models exhibit a slightly higher hallucination rate on the test set than on the training set. This could be attributed to the change in context, where the model is prompted with a question instead of a story title. Interestingly, OPT models in the unsegmented setting hallucinate more than T5 and Flan-T5 models on the test set, but not on the training set. This behavior might stem from OPT models overfitting training samples with positional embeddings, affecting their performance when prompted with questions, which differ in length from titles.

Conversely, for the segmented-trained models, hallucination rates among different models are more similar and also decrease with scale. However, whether this decline continues as models increase in size is uncertain. To elucidate this, experiments with larger models are essential.

### 4.4 Effect of Scale

Both key metrics we use to study knowledge consolidation: recall performance and hallucination rate, seem to improve as model size increases. However, given the improvement in performance in both the unsegmented and segmented settings, this is not conclusive evidence to knowledge consolidation happening with scale. To support the emergent behavior hypothesis (Wei et al., 2022b), the improvement rate in the segmented setting should significantly outpace that in the unsegmented one. Additionally, it remains uncertain if performance in the segmented scenario will eventually plateau, perhaps before reaching the performance levels of

models trained on unsegmented stories. To truly gauge the impact of scale on knowledge consolidation, experiments with larger models are needed, but we unfortunately lack the compute to run them.

## 5 Related Work

**Knowledge Representation:** Results from probing neural language models have shown models not only encoding facts (Petroni et al., 2019) or linguistic knowledge (Tenney et al., 2019) in their parameters, but also using them in downstream tasks (Peters et al., 2018; Goldberg, 2019; Kazemnejad et al., 2023). The amount of knowledge a model retains in the parameters (Dhingra et al., 2022; AlKhamissi et al., 2022; Roberts et al., 2020) is perceived as a reflection of the models' success in downstream tasks (Moghaddam and Honey, 2023). However, relying on parameters for knowledge has shown that language models can hallucinate (Ji et al., 2023) and struggle to model less frequent data (Kandpal et al., 2022). Going further than the existing work, with the proposed EpiK-Eval framework we attempt to understand LMs' behavior towards knowledge representation of segmented text chunks describing a set of relation-categories.

**Multi-Hop QA:** In multi-hop question answering (QA) benchmarks (Welbl et al., 2017; Yang et al., 2018; Ho et al., 2020; Mavi et al., 2022), models are tasked with answering questions by navigating through multiple documents and extracting relevant information from them. This process is pure inference, with the model relying on external knowledge sourced directly from the documents.

Conversely, we focus on investigating how well these models can recall and consolidate the knowledge already embedded within their parameter space—knowledge acquired during training (referred to as "internal knowledge"). This contrasts with merely assessing the model's ability to conduct document-based searches.

**Artifacts of Reasoning in LMs:** To utilize the stored knowledge, approaches such as prompting and in-context learning (Wei et al., 2022a,b,c; Liu et al., 2023) have gained popularity for tasks involving reasoning over a given context. While LMs have shown strong reasoning skills when information is fully available in the context (Han et al., 2022; Zelikman et al., 2022), inconsistent results appear when such is not the case (Gu et al., 2023). While Li et al. (2021) demonstrate that LMs

maintain state information, the authors probe for factual information that does not require consolidation. Unlike existing works, using EpiK-Eval, we focus on studying the effect of information spread during a LM's training on the model's ability to recall and consolidate the knowledge at inference.

## 6 Discussion

**Consolidating Knowledge in Language Models:** Our study delineates the limitations of language models in consolidating knowledge across different text sequences, compared to a noticeably stronger performance when working within a single text sequence. We attribute this disparity primarily to the core objective of such models: to enhance word prediction within given sequences, while also using knowledge from previously processed text sequences, encoded in the model's parameters.

Current pre-training objectives such as masked and causal language modeling (Devlin et al., 2019; Brown et al., 2020) potentially prioritize learning dependencies within text sequences over those spanning across multiple ones. For instance, a cause-and-effect relationship could exist between two sequences. However, if the content of the first does not explicitly help in predicting the second's content, the model might not learn this relation. Consequently, numerous inter-sequence dependencies in the training corpus, which may hold significant importance in downstream tasks, may be ignored owing to their perceived irrelevance in the next-word prediction task. In contrast, the model can readily establish correlational dependencies within individual sequences which can even lead to the direct memorization of text, a frequent occurrence in LLMs (Carlini et al., 2020; McCoy et al., 2021; Tirumala et al., 2022; Carlini et al., 2023).

In light of these arguments and results, we assert the need to revisit the training objectives of language models. To utilize these models effectively, we should prioritize devising training methods that capture and consolidate the numerous information dependencies within the training corpus. A potential avenue to explore could be to guide these models in consolidating their knowledge via methods such as RL-HF (Bai et al., 2022) or self-taught (Zelikman et al., 2022).

**Exploiting Longer Context vs Knowledge Consolidation:** In response to the knowledge consolidation challenge faced by LMs, it could be ar-

gued that the inclusion of a comprehensive context through prompts could be an effective alternative to having the LM remember the necessary context autonomously. This proposition is emboldened by recent successes in extending the context window size (Xiong et al., 2022; Ratner et al., 2023; Anthropic, 2023) as well as the sequence length (Dai et al., 2019; Gu et al., 2022; Poli et al., 2023; Bertsch et al., 2023). Such additional information can be supplied by either a user or an auxiliary system (Nakano et al., 2022; Schick et al., 2023; Patil et al., 2023; Paranjape et al., 2023).

Expecting humans to provide comprehensive context may, however, be impractical. Given the diverse range of specialist knowledge needed for various tasks, it's possible for a user to lack the necessary expertise. On the other hand, integrating auxiliary systems to provide these contexts presents a challenge analogous to that faced by LMs. To be useful, such an auxiliary system must understand and retain all relevant interdependencies within the training corpus related to problem-solving. Unfortunately, current auxiliary systems, such as search engines or retrieval tools (Karpukhin et al., 2020; Guu et al., 2020; Lewis et al., 2020), fall short of this holistic understanding and recall of context.

Another strategy leveraging longer context windows can be to train LMs on concatenated text sequences with inherent relevance (Shi et al., 2023). This approach, however, presents its own complexities. The innumerable ways texts can interrelate complicates the process of determining and training on all possible combinations. Hence current solutions do not provide a comprehensive solution to this issue.

**Knowledge Consolidation at Scale:** Our study underscores a substantial discrepancy in performance between models trained on segmented stories and those trained on unsegmented stories. If we assume that the recall performance for models in the segmented setting continues to improve without plateauing prematurely, our estimates (Caballero et al., 2022) suggest that a model with 172B parameters, trained on our benchmark's segmented stories, would be required to match the performance of an 80M parameter model trained on the unsegmented stories.

Although consolidating knowledge from fragmented text sequences arguably poses a greater challenge than from a singular cohesive text, the margin for enhancement in this domain is possibly significant. As we venture into the realm of real-world applications (OpenAI, 2023; Anil et al., 2023; Touvron et al., 2023), there exist a wide array of settings that necessitate a LLM to recall and integrate data from multiple text sequences. Accordingly, enhancing this ability can potentially elevate the efficiency, robustness and performance of such models, thereby redefining the landscape of complex language tasks.

One challenge with studying this problem at scale is distinguishing whether LLMs demonstrate an improved ability to model dependencies within their training corpus (emergent behavior) or if the dataset diversity enables the extraction of most dependencies of interest within single text sequences in the corpus. To probe for knowledge consolidation at scale, we propose the use of self-contained narratives such as short stories or books. These documents can be segmented and dispersed within the training corpus of LLMs (Touvron et al., 2023; Computer, 2023) and evaluation can be performed in a similar fashion as EpiK-Eval, where questions can assess the understanding of the overall narrative and the various relations in the story. With complex enough naratives, this methodology should provide a robust framework for examining the knowledge consolidation capabilities of LLMs.

# 7 Conclusion

In this paper, we presented the EpiK-Eval benchmark, a tool designed specifically to evaluate the proficiency of LMs in consolidating their knowledge for problem-solving tasks. Our findings underscore the limitations of current LMs, which appear to mostly maintain a disjoint knowledge state of training observations. Further, we note a significant performance gap and an increased rate of hallucinations for models trained on segmented narratives compared to those trained on unsegmented ones. We attribute these discrepancies to the training objectives of the models, which underscores the need to more effectively model the dependencies within the training corpus. By highlighting current limitations and opportunities for improving LMs, these results delineate paths for future research, hopefully enabling the growth of language models beyond simple knowledge bases.

## Limitations

Ensuring that EpiK-Eval's data doesn't leak into the pre-training set of LLMs is a challenge. This

inclusion could skew the benchmark's results. One straightforward solution is to check if the data exists within the pre-training set, though this method is computationally intensive. Another practical approach is to generate and release a new version of the dataset periodically, for instance, annually. To further safeguard against potential leaks, we've encrypted the data in the public release of the benchmark. Users are required to decrypt it locally before use.

## Ethics Statement

This study employs machine learning algorithms, specifically large language models, which are trained on vast amounts of text data. While these models have shown remarkable predictive capabilities, it is important to underscore the ethical concern that arises from their training process. These models often learn from data that is intrinsically embedded with human biases, which can subsequently be reflected in their outputs. Therefore, it is paramount to approach any output produced by these models with critical consideration of this potential for embedded bias.

## Acknowledgements

Jerry Huang is supported by a Natural Sciences and Engineering Research Council of Canada (NSERC) Canada Graduate Scholarship and Fonds de Recherche du Québec Nature et technologies (FRQNT) training scholarship.

Sarath Chandar is supported by the Canada CIFAR AI Chairs program, the Canada Research Chair in Lifelong Machine Learning, and the NSERC Discovery Grant.

This research was enabled in part by compute resources provided by Mila (mila.quebec) and the Digital Research Alliance of Canada (alliance-can.ca).

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

## A Training Details

**T5 & Flan-T5:** All models are fine-tuned for 360,000 steps with a batch size of 50. We use the Adam optimizer, setting a base learning rate of $1 \times 10^{-4}$. The learning rate undergoes a linear warmup for the initial 1% of training steps, after which it remains constant. No weight decay or gradient clipping is applied.

**OPT:** Except for the learning rate, we use the same hyperparameters as with T5 and Flan-T5. The base learning rates for different OPT model sizes are:

- 125M: $6 \times 10^{-5}$

- 350M: $3 \times 10^{-5}$

- 1.3B: $2 \times 10^{-5}$

- 2.7B: $1.6 \times 10^{-5}$

## B Per Task Description & Results

We provide a detailed description of each task in Tables 3-20, along with the per task results in Figures 5-22.

## C Hallucination Examples

In Table 21 and Table 22, we present examples of hallucinations observed in models trained on segmented stories. Our analysis revealed no significant differences in the patterns of hallucinations across various models. It's also worth noting that models trained on unsegmented stories exhibited similar hallucination patterns, albeit at a reduced frequency (as shown in Figure 4).

## D Recall Length Distribution

We analyzed the length of story recalls in relation to the target distribution to determine the impact of training on segmented versus unsegmented stories. Figure 23 displays the distribution of the recall length, measured in number of sentences, for both the model and the target. For brevity, we present results only for the largest variant of each model, noting that similar patterns were observed across all model sizes. Our analysis revealed no significant differences between these distributions, leading us to conclude that training on segmented stories does not influence the recall length of the models' outputs.

## Task 1: "List the different *x*."

**Category:** Listing
**Description:** The objective of this task is to identify and list the days on which a person worked from home.

| Template | Example |
|---|---|
| **Story:** | **Story:** |
| [Task 1] {*name*}'s Work From Home Log | [Task 1] Tom's Work From Home Log |
| {*name*} worked from home on {*day*}. | Tom worked from home on Monday. |
| ... | Tom worked from home on Friday. |
| {*name*} worked from home on {*day*}. | |
| **Question:** | **Question:** |
| [Task 1] Which days did {*name*} work from home? | [Task 1] Which days did Tom work from home? |
| **Answer:** | **Answer:** |
| {*story*} | Tom worked from home on Monday. |
| The answer is {*answer*}. | Tom worked from home on Friday. |
| | The answer is Monday and Friday. |

**Details**

- {*day*}: Randomly sampled without replacement from ["Monday", "Wednesday", "Friday"].
- {*answer*}: Comprises the days listed.
- The story can span one to three sentences, excluding the title. Sentences are ordered chronologically based on {*day*}.

Table 3: Templates for generating Task 1 stories, questions, and answers, with an example provided.

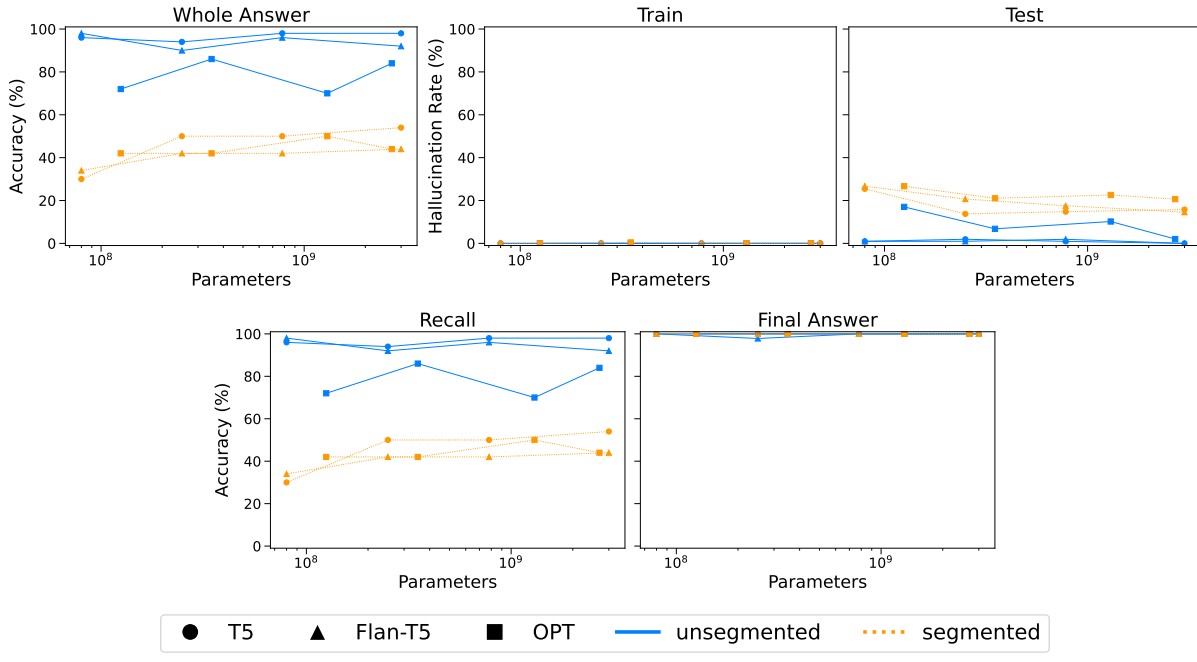

Figure 5: Task 1 results. Top left: percentage of correct answers. Top right: hallucination rate for both train and test sets. Bottom: percentage of correct recalls (left) and final answers (right).

**Task 2: "How many times does *x* happen?"**

**Category:** Counting
**Description:** The task aims to count the number of times the fishing activity occurred within the story.

| Template | Example |
|---|---|
| **Story:**
[Task 2] {*name*}'s Vacation
{*name*} went {*activity*} on {*day*}.
...
{*name*} went {*activity*} on {*day*}. | **Story:**
[Task 2] Tom's Vacation
Tom went fishing on Monday.
Tom went hiking on Wednesday.
Tom went fishing on Thursday.
Tom went hiking on Saturday.
Tom went hiking on Sunday. |
| **Question:**
[Task 2] How many times did {*name*} go fishing? | **Question:**
[Task 2] How many times did Tom go fishing? |
| **Answer:**
{*story*}
The answer is {*answer*}. | **Answer:**
Tom went fishing on Monday.
Tom went hiking on Wednesday.
Tom went fishing on Thursday.
Tom went hiking on Saturday.
Tom went hiking on Sunday.
The answer is 2. |

**Details**

- {*activity*}: Randomly sampled with replacement from ["fishing", "hiking"].
- {*day*}: Randomly sampled without replacement from the seven days of the week.
- {*answer*}: A numeric value representing the count.
- The story comprises 3 to 5 sentences, excluding the title. Sentences are ordered chronologically by {*day*}.

Table 4: Templates for generating Task 2 stories, questions, and answers, with an example provided.

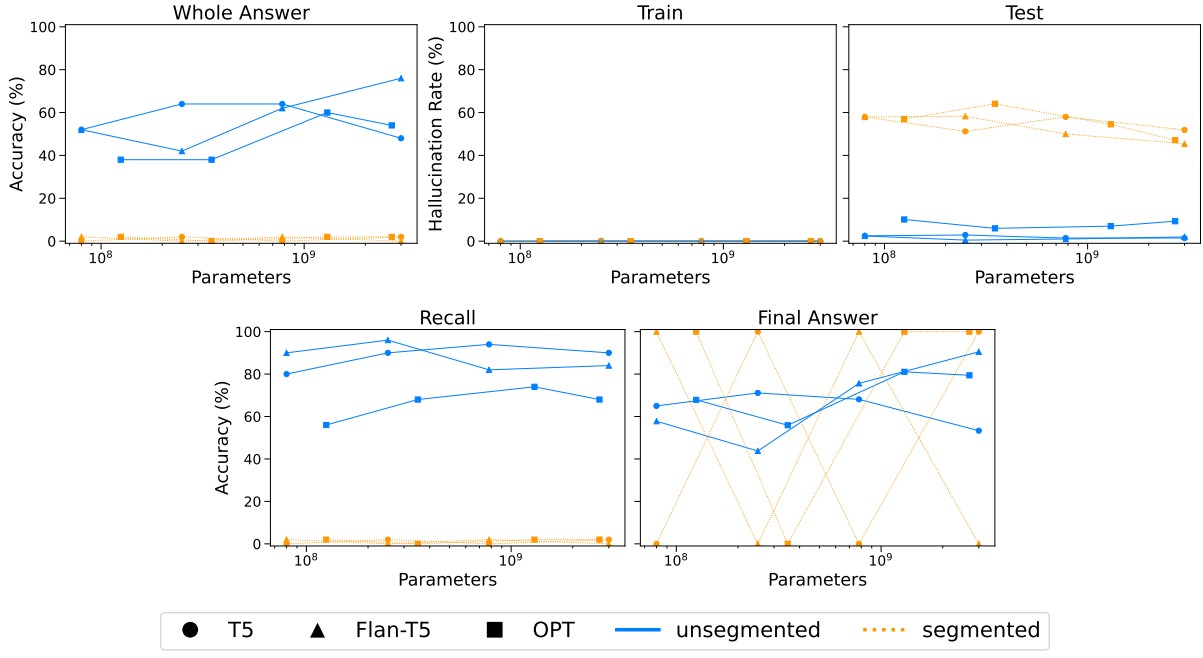

Figure 6: Task 2 results. Top left: percentage of correct answers. Top right: hallucination rate for both train and test sets. Bottom: percentage of correct recalls (left) and final answers (right).

**Task 3: "Does *x* happen more/less often than *y*?"**

**Category:** Ranking
**Description:** The objective of this task is to determine whether the person has more meetings with Person A or Person B.

| Template | Example |
|---|---|
| **Story:**
[Task 3] {*name*}'s Afternoon
{*time*} - {*name*} {*activity*}.
...
{*time*} - {*name*} {*activity*}. | **Story:**
[Task 3] Tom's Afternoon
1:00 PM - Tom has a meeting with co-worker A.
2:00 PM - Tom fills up some forms.
3:00 PM - Tom has a meeting with co-worker B.
4:00 PM - Tom fills up some forms.
5:00 PM - Tom has a meeting with co-worker A. |
| **Question:**
[Task 3] Does {*name*} have more meetings with co-worker A or B? | **Question:**
[Task 3] Does Tom have more meetings with co-worker A or B? |
| **Answer:**
{*story*}
The answer is {*answer*}. | **Answer:**
1:00 PM - Tom has a meeting with co-worker A.
2:00 PM - Tom fills up some forms.
3:00 PM - Tom has a meeting with co-worker B.
4:00 PM - Tom fills up some forms.
5:00 PM - Tom has a meeting with co-worker A.
The answer is A. |

**Details**

- {*time*}: Randomly sampled without replacement from ["1:00 PM", "2:00 PM", "3:00 PM", "4:00 PM", "5:00 PM"].
- {*activity*}: Randomly sampled with replacement from ["has a meeting with co-worker A", "has a meeting with co-worker B", "fills up some forms"].
- {*answer*}: Either "A" or "B", based on the frequency of the meetings.
- The story consists of 3 to 5 sentences, excluding the title. Sentences are chronologically ordered by {*time*}.

Table 5: Templates for generating Task 3 stories, questions, and answers, with an example provided.

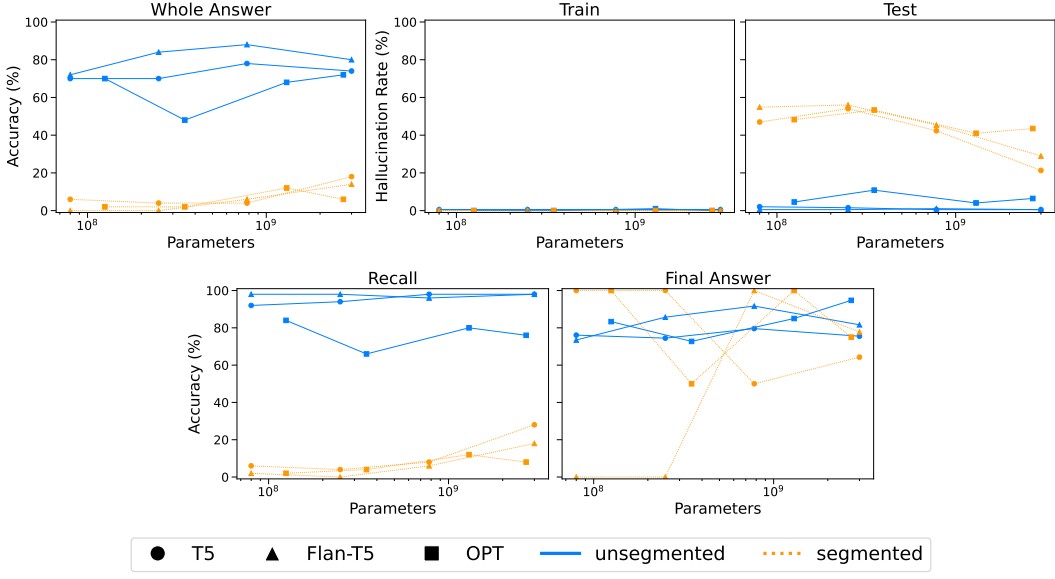

Figure 7: Task 3 results. Top left: percentage of correct answers. Top right: hallucination rate for both train and test sets. Bottom: percentage of correct recalls (left) and final answers (right).

**Task 4: "Does *x* happen before/after *y*?"**

**Category:** Temporal

**Description:** The task is designed to ascertain whether a specific event happened before or after another event. Additionally, a reasoning based on the order of months is provided to justify the answer.

| Template | Example |
|---|---|
| **Story:** | **Story:** |
| [Task 4] {*name*}'s Year | [Task 4] Tom's Year |
| {*name*} {*event*} in {*month*}. | Tom buys a house in March. |
| ... | Tom goes on a vacation in June. |
| {*name*} {*event*} in {*month*}. | Tom gets married in October. |
| **Question:** | **Question:** |
| [Task 4] Does {*name*} {*event_a*} {*before/after*} they {*event_b*}? | [Task 4] Does Tom buy a house after they get married? |
| **Answer:** | **Answer:** |
| {*story*} | Tom buys a house in March. |
| {*month_a*} is {*reasoning*} {*month_b*}. | Tom goes on a vacation in June. |
| The answer is {*answer*}. | Tom gets married in October. |
|  | March is not after October. |
|  | The answer is no. |

**Details**

- {*event*}: Randomly sampled without replacement from ["buys a house", "goes on a vacation", "gets married"].
- {*month*}: Randomly sampled without replacement from ["January", "March", "June", "August", "October"].
- {*event_a*} and {*event_b*} are randomly drawn among the sampled {*event*}.
- {*month_a*} and {*month_b*} are associated with the corresponding {*event_a*} and {*event_b*}.
- {*before/after*}: Randomly sampled between "before" and "after".
- {*reasoning*}: Explains the temporal relationship between {*month_a*} and {*month_b*}. Options include "before", "after", "not before", and "not after".
- {*answer*}: A simple "yes" or "no".
- The story consists of 2 to 3 sentences, excluding the title. Sentences are chronologically ordered by {*month*}.

Table 6: Templates for generating Task 4 stories, questions, and answers, with an example provided.

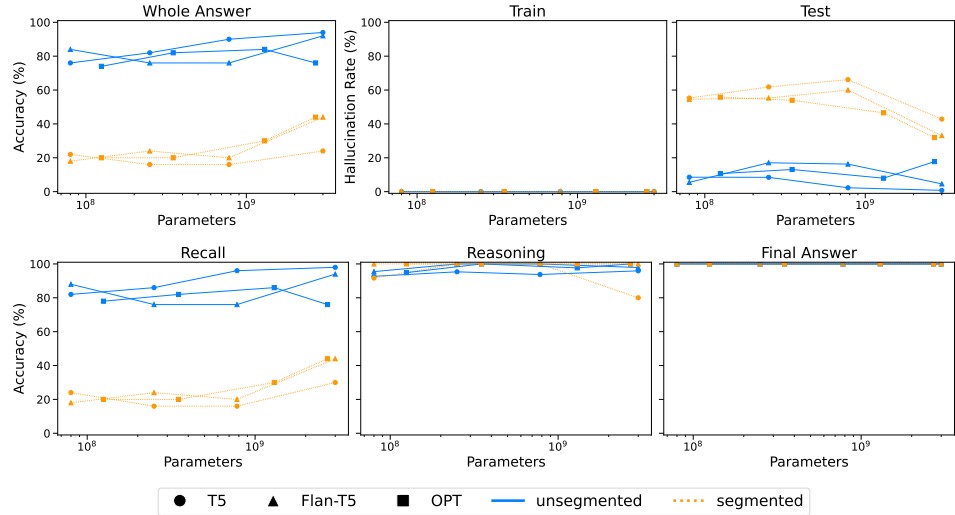

Figure 8: Task 4 results. Top left: percentage of correct answers. Top right: hallucination rate for both train and test sets. Bottom: percentage of correct recalls (left), reasoning (center), and final answers (right).

**Task 5: "When *x* happens, does *y* happen?"**

**Category:** Temporal
**Description:** This task aims to determine whether, on days when person A is in one specific location, person B is in another specific location.

| Template | Example |
|---|---|
| **Story:** | **Story:** |
| [Task 5] {*name_a*} and {*name_b*}'s Travel Log | [Task 5] Tom and Alice's Travel Log |
| {*name_a*} was in {*location*} on {*day*}. | Tom was in Paris on Monday. |
| ... | Tom was in New York on Tuesday. |
| {*name_a*} was in {*location*} on {*day*}. | Alice was in Los Angeles on Monday. |
| {*name_b*} was in {*location*} on {*day*}. | Alice was in Rome on Tuesday. |
| ... | |
| {*name_b*} was in {*location*} on {*day*}. | |
| **Question:** | **Question:** |
| [Task 5] When {*name_a*} is in {*location_a*}, is {*name_b*} in {*location_b*}? | [Task 5] When Tom is in Paris, is Alice in Rome? |
| **Answer:** | **Answer:** |
| {*story*} | Tom was in Paris on Monday. |
| Those are {*reasoning*} days. | Tom was in New York on Tuesday. |
| The answer is {*answer*}. | Alice was in Los Angeles on Monday. |
| | Alice was in Rome on Tuesday. |
| | Those are different days. |
| | The answer is no. |

**Details**

- {*location*} for person A is chosen without replacement from ["Paris", "New York", "Vancouver"], and for person B from ["Los Angeles", "Rome", "Tokyo"].
- {*day*} is picked without replacement from ["Monday", "Tuesday", "Wednesday"].
- {*location_a*} and {*location_b*} are randomly drawn from the sampled {*location*} for person A and B respectively.
- {*reasoning*}: Specifies whether the days of the events in question are "the same" or "different".
- {*answer*}: A simple "yes" or "no".
- Each person's events are ordered by {*day*}—person A's events are listed first, followed by person B's events. There can be between 2 and 3 sentences per person.

Table 7: Templates for generating Task 5 stories, questions, and answers, with an example provided.

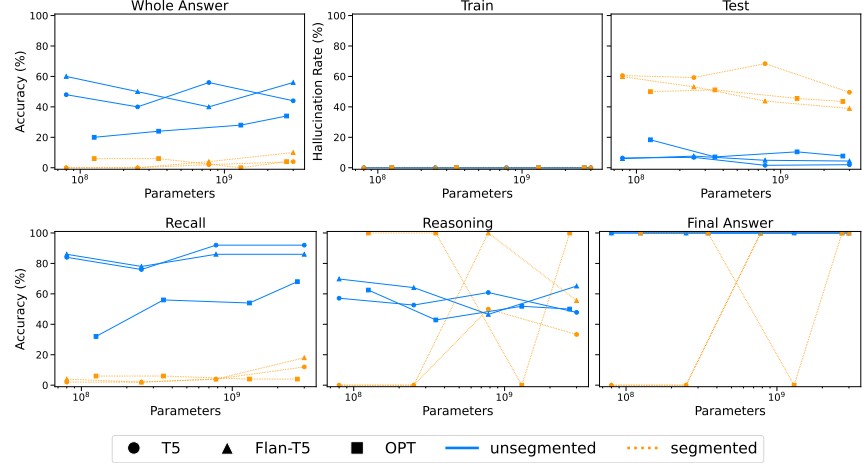

Figure 9: Task 5 results. Top left: percentage of correct answers. Top right: hallucination rate for both train and test sets. Bottom: percentage of correct recalls (left), reasoning (center), and final answers (right).

## Task 6: "Is *x* the only time that *y* happens?"

**Category:** Uniqueness
**Description:** Determine whether a person engaged in a specific activity only once during the week.

| Template | Example |
|---|---|
| **Story:**
[Task 6] {*name*}'s Holiday
{*name*} {*activity*} on {*day*}.
...
{*name*} {*activity*} on {*day*}. | **Story:**
[Task 6] Tom's Holiday
Tom goes hiking on Monday.
Tom goes fishing on Tuesday.
Tom goes to the park on Wednesday.
Tom plays golf on Thursday.
Tom visits a friend on Friday. |
| **Question:**
[Task 6] {*name*} {*activity_a*} on {*day_a*}. Is it the only time that week that {*name*} {*activity_a*}? | **Question:**
[Task 6] Tom goes fishing on Tuesday. Is it the only time that week that Tom goes fishing? |
| **Answer:**
{*story*}
The anwer is {*answer*}. | **Answer:**
Tom goes hiking on Monday.
Tom goes fishing on Tuesday.
Tom goes to the park on Wednesday.
Tom plays golf on Thursday.
Tom visits a friend on Friday.
The answer is yes. |

### Details

- {*activity*} is chosen with replacement from ["goes hiking", "goes fishing", "goes to the park", "plays golf", "visits a friend"].
- {*day*} is picked without replacement from ["Monday", "Tuesday", "Wednesday", "Thursday", "Friday"].
- {*activity_a*} is a randomly selected {*activity*}, and {*day_a*} is its corresponding day.
- {*answer*} can be "yes" or "no".
- The story contains 4 to 5 sentences, excluding the title. Sentences are ordered by {*day*}.

Table 8: Templates for generating Task 6 stories, questions, and answers, with an example provided.

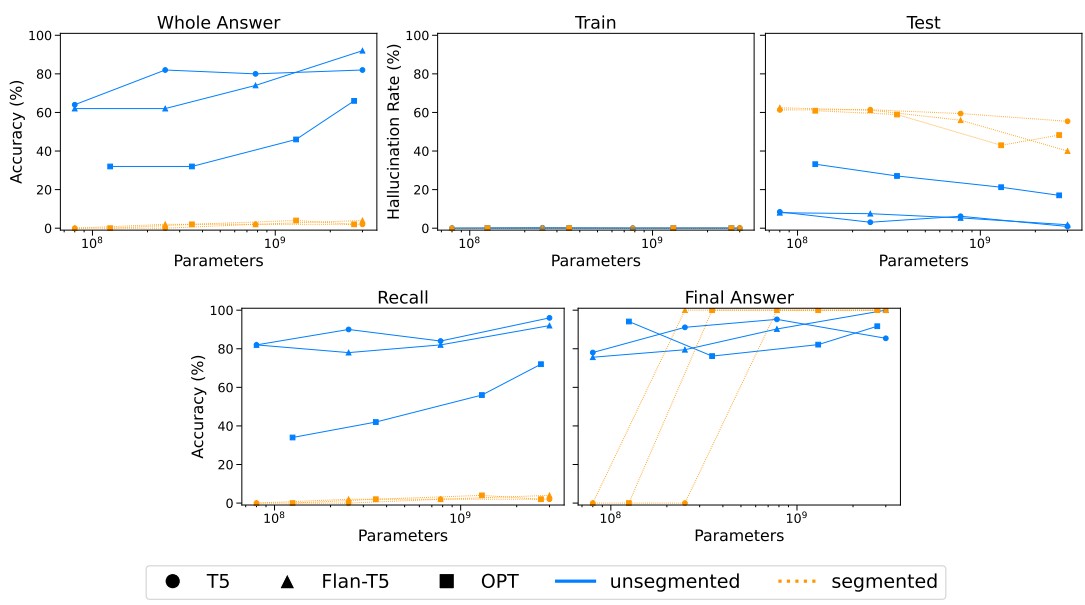

Figure 10: Task 6 results. Top left: percentage of correct answers. Top right: hallucination rate for both train and test sets. Bottom: percentage of correct recalls (left) and final answers (right).

## Task 7: "Between *x* and *y*, does *z* happen?"

**Category:** Temporal

**Description:** Determine if a person performs a specific activity between two other distinct activities during the day.

| Template | Example |
|---|---|
| **Story:**
[Task 7] {*name*}'s Day
{*time*}, {*name*} {*activity*}.
...
{*time*}, {*name*} {*activity*}. | **Story:**
[Task 7] Tom's Day
Morning, Tom goes for a walk.
Noon, Tom makes a phone call.
Afternoon, Tom makes tea.
Evening, Tom reads a book. |
| **Question:**
[Task 7] Between {*activity_a*} and {*activity_b*}, does {*name*} {*activity_c*}? | **Question:**
[Task 7] Between going for a walk and making tea, does Tom read a book? |
| **Answer:**
{*story*}
The answer is {*answer*}. | **Answer:**
Morning, Tom goes for a walk.
Noon, Tom makes a phone call.
Afternoon, Tom makes tea.
Evening, Tom reads a book.
The answer is no. |

### Details

- {*activity*} is chosen without replacement from ["goes for a walk", "makes a phone call", "makes tea", "reads a book"].
- {*time*} is chosen without replacement from the sequential list ["Morning", "Noon", "Afternoon", "Evening"].
- {*activity_a*} and {*activity_b*} are randomly selected among the sampled {*activity*}.
- {*activity_c*} is sampled from the list of activities but cannot be the same as {*activity_a*} or {*activity_b*}.
- {*answer*} is either "yes" or "no".
- The story contains 3 to 4 sentences, excluding the title, with sentences ordered chronologically by {*time*}.

Table 9: Templates for generating Task 7 stories, questions, and answers, with an example provided.

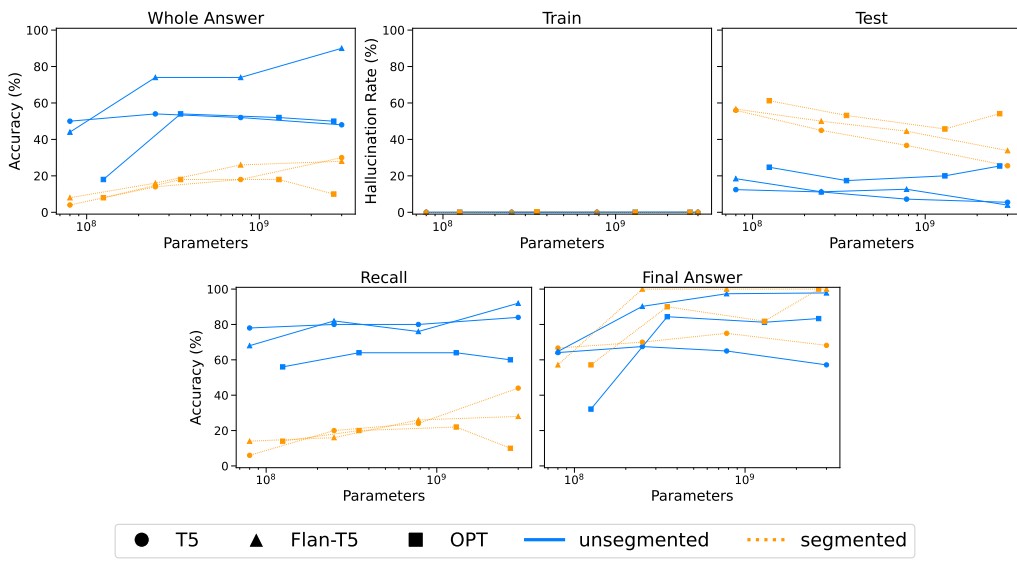

Figure 11: Task 7 results. Top left: percentage of correct answers. Top right: hallucination rate for both train and test sets. Bottom: percentage of correct recalls (left) and final answers (right).

**Task 8: "How much time has passed between *x* and *y*?"**

**Category:** Temporal
**Description:** Determine the duration in hours between two activities a person engaged in.

| Template | Example |
|---|---|
| **Story:**
[Task 8] {*name*}'s Contact Log
At {*time*}, {*name*} {*event*}.
...
At {*time*}, {*name*} {*event*}. | **Story:**
[Task 8] Tom's Contact Log
At 2pm, Tom wrote a letter.
At 4pm, Tom sent an email.
At 7pm, Tom made a phone call. |
| **Question:**
[Task 8] How much time passed between {*name*} {*event_a*} and {*event_b*}? | **Question:**
[Task 8] How much time passed between Tom wrote a letter and sent an email? |
| **Answer:**
{*story*}
{*reasoning*}
The answer is {*answer*}. | **Answer:**
At 2pm, Tom wrote a letter.
At 4pm, Tom sent an email.
At 7pm, Tom made a phone call.
4 - 2 = 2.
The answer is 2. |

**Details**

- {*time*} is selected without replacement from ["1pm", "2pm", "3pm", "4pm", "5pm"].
- {*event*} is selected without replacement from ["wrote a letter", "sent an email", "made a phone call", "started a video chat"].
- {*event_a*} and {*event_b*} are randomly chosen among the sampled {*event*}, with {*event_b*} always occurring after {*event_a*}.
- {*reasoning*} describes the subtraction of the times corresponding to {*event_a*} from {*event_b*}, representing the duration in hours.
- {*answer*} indicates the number of hours.
- The story contains 3 to 4 sentences, excluding the title, arranged chronologically by {*time*}.

Table 10: Templates for generating Task 8 stories, questions, and answers, with an example provided.

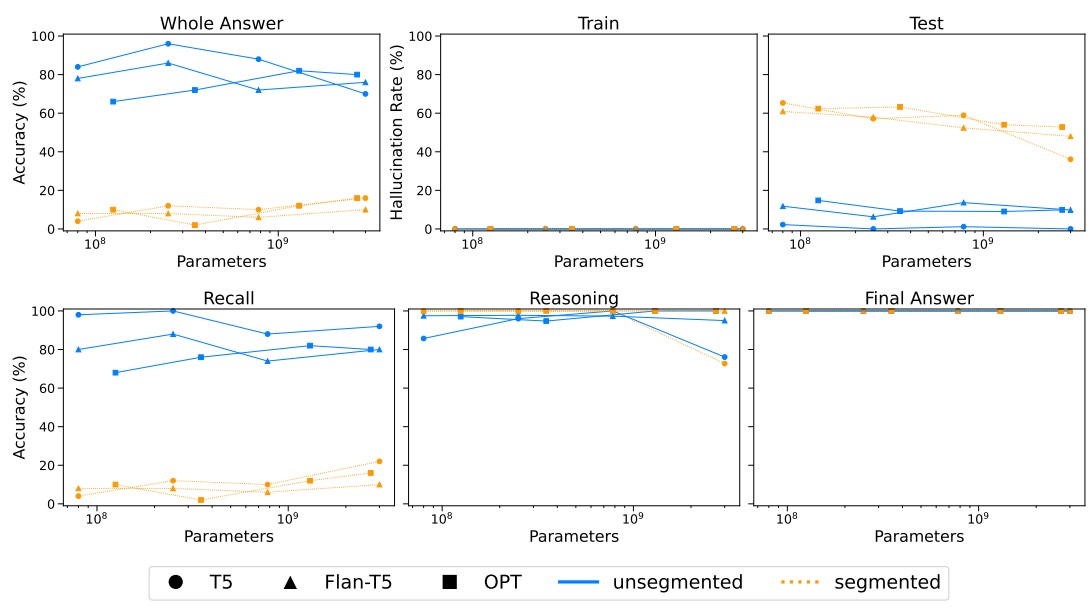

Figure 12: Task 8 results. Top left: percentage of correct answers. Top right: hallucination rate for both train and test sets. Bottom: percentage of correct recalls (left), reasoning (center), and final answers (right).

## Task 9: "At what time does *y* happen based on *x*?"

**Category:** Temporal
**Description:** Determine the time the person asks for the bill based on prior events at the restaurant.

| Template | Example |
|---|---|
| **Story:** | **Story:** |
| [Task 9] {*name*} at the Restaurant | [Task 9] Tom at the Restaurant |
| {*name*} arrived at the restaurant at {*time*}. | Tom arrived at the restaurant at 6:00 PM. |
| {*minute*} minutes after arriving, {*name*} ordered a {*item_1*}. | 2 minutes after arriving, Tom ordered a drink. |
| {*minute*} after ordering a {*item_1*}, {*name*} ordered a {*item_2*}. | 1 minutes after ordering a drink, Tom ordered a hamburger. |
| {*minute*} minutes ordering a {*item_2*}, {*name*} asked for the bill. | 3 minutes after ordering a hamburger, Tom asked for the bill. |
| **Question:** | **Question:** |
| [Task 9] At what time does {*name*} ask for the bill? | [Task 9] At what time does Tom ask for the bill? |
| **Answer:** | **Answer:** |
| {*story*} | Tom arrived at the restaurant at 6:00 PM. |
| {*reasoning*} | 2 minutes after arriving, Tom ordered a drink. |
| The answer is {*answer*}. | 1 minutes after ordering a drink, Tom ordered a hamburger. |
|  | 3 minutes after ordering a hamburger, Tom asked for the bill. |
|  | {*reasoning*} |
|  | The answer is {*answer*}. |

**Details**

- {*time*} is selected between "6:00 PM" and "6:30 PM," rounded to the nearest minute.
- {*minute*} is chosen with replacement from ["1", "2", "3"].
- {*item_1*} can be either "drink" or "coffee".
- {*item_2*} can be "hamburger" or "sandwich".
- There's a 50% chance {*name*} won't order {*item_2*}. If so, the penultimate sentence is omitted, and the last sentence references {*item_1*}.
- {*reasoning*} provides the total time elapsed from the arrival to the request for the bill.
- {*answer*} indicates the exact time.
- The story contains 3 or 4 sentences, not counting the title, depending on whether or not {*item_2*} was ordered.

Table 11: Templates for generating Task 9 stories, questions, and answers, with an example provided.

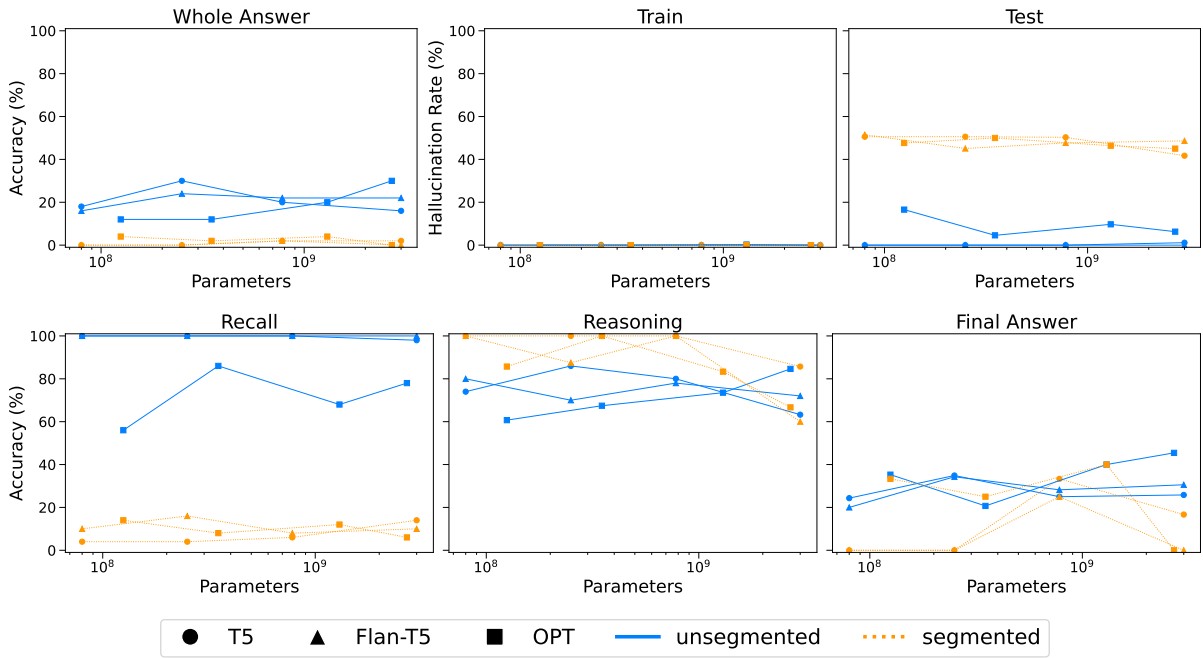

Figure 13: Task 9 results. Top left: percentage of correct answers. Top right: hallucination rate for both train and test sets. Bottom: percentage of correct recalls (left), reasoning (center), and final answers (right).

**Task 10: "The *x*'th time that *y* happens, what is a unique detail about *y* compared to the other *x* times?"**

**Category:** Uniqueness

**Description:** Determine who accompanied the person the *x*'th time they engaged in a specific activity.

| Template | Example |
|---|---|
| **Story:** | **Story:** |
| [Task 10] {*name*} Hunting and Canoeing Week | [Task 10] Tom's Hunting and Canoeing Week |
| {*day*}, {*name*} went {*activity*} with {*friend*}. | Monday, Tom went hunting with Alice. |
| … | Tuesday, Tom went canoeing with Bob. |
| {*day*}, {*name*} went {*activity*} with {*friend*}. | Wednesday, Tom went hunting with Carl. |
| | Thursday, Tom went canoeing with James. |
| | Friday, Tom went canoeing with Steve. |
| **Question:** | **Question:** |
| [Task 10] The {*x*} time that {*name*} went | [Task 10] The second time that Tom went hunting, |
| {*q_activity*}, who else was there? | who else was there? |
| **Answer:** | **Answer:** |
| {*story*} | Monday, Tom went hunting with Alice. |
| The answer is {*answer*}. | Tuesday, Tom went canoeing with Bob. |
| | Wednesday, Tom went hunting with Carl. |
| | Thursday, Tom went canoeing with James. |
| | Friday, Tom went canoeing with Steve. |
| | The answer is Carl. |

**Details**

- {*day*} can be any day of the week.
- {*activity*} in each statement can be either "canoeing" or "hiking". However, "hunting" must be picked at least twice but no more than three times.
- {*friend*} is randomly sampled from a list of names.
- {*q_activity*} can be either "canoeing" or "hiking".
- {*x*} is a number between 1 and the number of times {*q_activity*} occurs.
- {*answer*} is the name of the person who was with {*name*} during the {*x*}'th occurrence of the {*q_activity*}.
- The story comprises 4 or 5 sentences, not including the title. Sentences are arranged by {*day*}.

Table 12: Templates for generating Task 10 stories, questions, and answers, with an example provided.

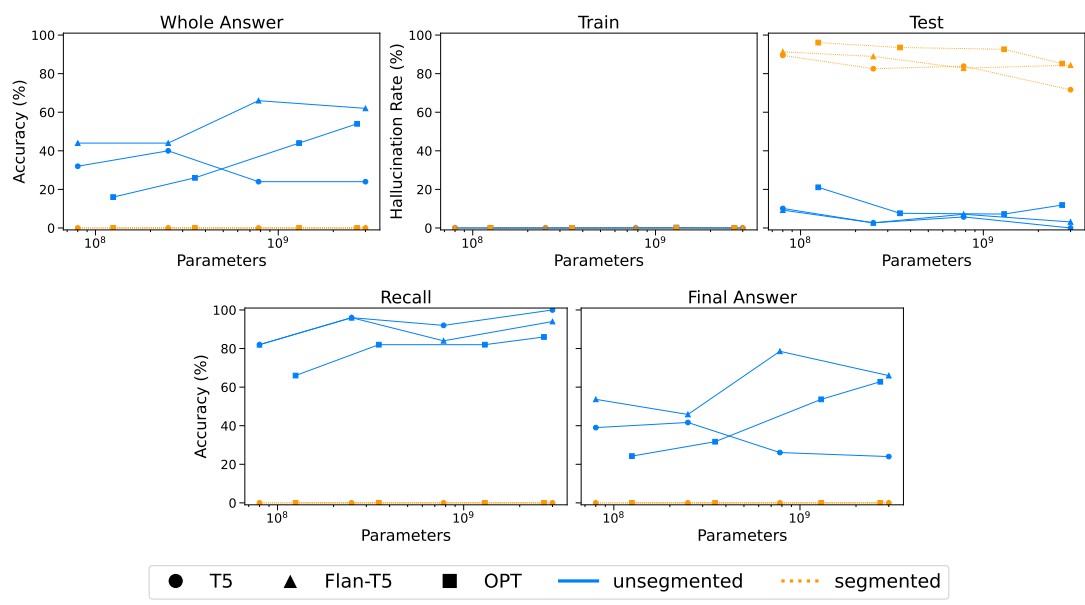

Figure 14: Task 10 results. Top left: percentage of correct answers. Top right: hallucination rate for both train and test sets. Bottom: percentage of correct recalls (left) and final answers (right).

## Task 11: "Every time *x* happens, is *y* always the same?"

**Category:** Consistency
**Description:** Determine if, every time a person travels to a specific location, they consistently use the same type of vehicle.

| Template | Example |
|---|---|
| **Story:**
[Task 11] {*name*}'s Car Choice
{*day*}, {*name*} drives to {*place*} in a {*vehicle*}.
. . .
{*day*}, {*name*} drives to {*place*} in a {*vehicle*}. | **Story:**
[Task 11] Tom's Car Choice
Monday, Tom drives to the grocery store in a minivan.
Tuesday, Tom drives to the pharmacy in a minivan.
Wednesday, Tom drives to the grocery store in a SUV.
Thursday, Tom drives to the grocery store in a SUV. |
| **Question:**
[Task 11] Every time {*name*} drives to {*q_place*}, is it always in a {*q_vehicle*}? | **Question:**
[Task 11] Every time Tom drives to the grocery store, is it always in a minivan? |
| **Answer:**
{*story*}
The answer is {*answer*}. | **Answer:**
Monday, Tom drives to the grocery store in a minivan.
Tuesday, Tom drives to the pharmacy in a minivan.
Wednesday, Tom drives to the grocery store in a SUV.
Thursday, Tom drives to the grocery store in a SUV.
The answer is no. |

### Details

- {*day*} is selected without replacement from ["Monday", "Tuesday", "Wednesday", "Thursday"].
- {*place*} in each statement can be either "the pharmacy" or "the grocery store".
- {*q_place*} is randomly chosen from the sampled {*place*}.
- {*vehicle*} in each statement can be either "minivan" or "SUV".
- {*q_vehicle*} can be either "minivan" or "SUV".
- {*answer*} is either "yes" or "no".
- The story consists of 3 to 4 sentences, excluding the title, and sentences are ordered by {*day*}.

Table 13: Templates for generating Task 11 stories, questions, and answers, with an example provided.

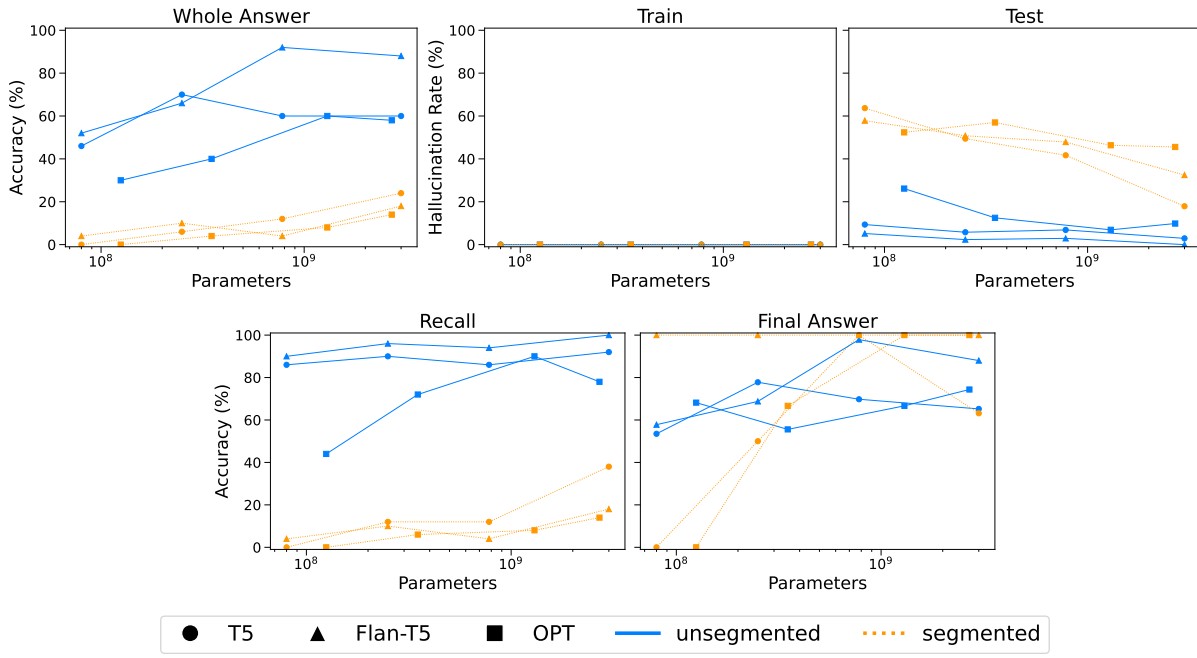

Figure 15: Task 11 results. Top left: percentage of correct answers. Top right: hallucination rate for both train and test sets. Bottom: percentage of correct recalls (left) and final answers (right).

**Task 12: "After how many *x* does *y* happen?"**

**Category:** Temporal
**Description:** The goal of the task is to determine after how many days person B joins person A.

| Template | Example |
|---|---|
| **Story:** | **Story:** |
| [Task 12] {*name*}'s Company | [Task 12] Tom's Company |
| {*day*}, {*name*} is alone. | Monday, Tom is alone. |
| ... | Tuesday, Tom is alone. |
| {*day*}, {*company*} arrives. | Wednesday, Alice arrives. |
| {*day*}, {*name*} is with {*company*}. | Thursday, Tom is with Alice. |
| ... | |
| {*day*}, {*name*} is with {*company*}. | |
| **Question:** | **Question:** |
| [Task 12] After how many days does {*company*} join {*name*}? | [Task 12] After how many days does Alice join Tom? |
| **Answer:** | **Answer:** |
| {*story*} | Monday, Tom is alone. |
| The answer is {*answer*}. | Tuesday, Tom is alone. |
| | Wednesday, Alice arrives. |
| | Thursday, Tom is with Alice. |
| | The answer is 2. |

**Details**

- {*day*} is sampled without replacement from ["Monday", "Tuesday", "Wednesday", "Thursday"].
- {*company*} is a randomly sampled name that is the same between statements.
- {*answer*} is a numeral indicating the number of days {*name*} was alone before being joined by {*company*}.
- The story comprises 3 to 4 sentences, excluding the title, and sentences are ordered chronologically by {*day*}.

Table 14: Templates for generating Task 12 stories, questions, and answers, with an example provided.

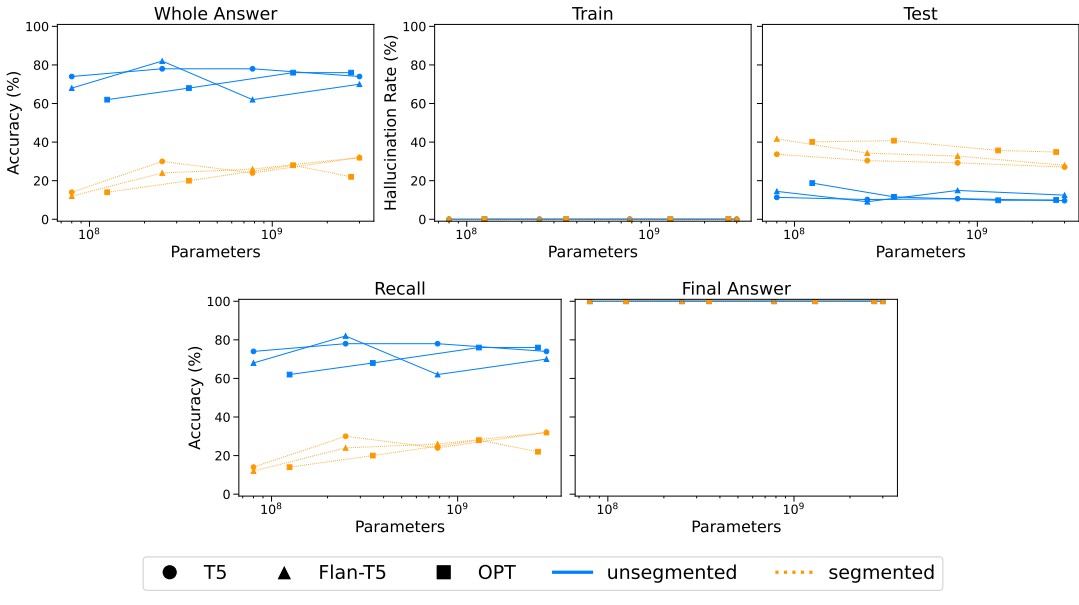

Figure 16: Task 12 results. Top left: percentage of correct answers. Top right: hallucination rate for both train and test sets. Bottom: percentage of correct recalls (left) and final answers (right).



**Task 13: "Is _x_ the _y_'th in the list?"**



| | |
|---|---|
| **Category:** Listing | |
| **Description:** The goal of the task is to determine if person B is the x'th person that person A meets. | |



| **Template** | **Example** |
|---|---|



| **Template** | **Example** |
|---|---|
| **Story:** | **Story:** |
| [Task 13] {_name_}'s Friends | [Task 13] Tom's Friends |
| {_name_} meets {_friend_} in the morning. | Tom meets Eve in the morning. |
| {_name_} meets {_friend_} at noon. | Tom meets Alice at noon. |
| {_name_} meets {_friend_} in the afternoon. | Tom meets Bob in the afternoon. |
| {_name_} meets {_friend_} in the evening. | |
| **Question:** | **Question:** |
| [Task 13] Is {_q_friend_} the {_x_} person that {_name_} meets? | [Task 13] Is Bob the second person that Tom meets? |
| **Answer:** | **Answer:** |
| {_story_} | Tom meets Eve in the morning. |
| {_q_friend_} is the {_reasoning_}. | Tom meets Alice at noon. |
| The answer is {_answer_}. | Tom meets Bob in the afternoon. |
| | Bob is the third. |
| | The answer is no. |



**Details**



- {_friend_} is a randomly sampled name.
- {_q_friend_} is randomly selected from one of the friends that {_name_} meets.
- {_x_} is randomly sampled from ["first", "second", "third", "fourth"].
- {_reasoning_} indicates the actual position of {_q_friend_} with respect to ["first", "second", "third", "fourth"].
- {_answer_} is either "yes" or "no".
- The story can have between 3 and 4 sentences, excluding the title. If there are only 3 sentences, the last one "{_name_} meets {_friend_} in the evening." is omitted.



Table 15: Templates for generating Task 13 stories, questions, and answers, with an example provided.



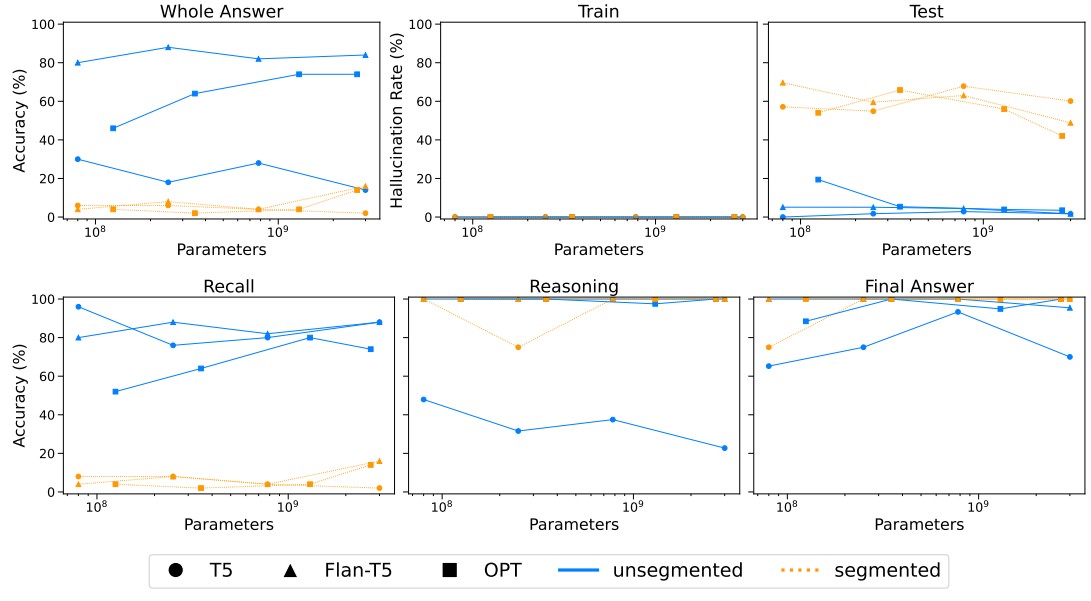

Figure 17: Task 13 results. Top left: percentage of correct answers. Top right: hallucination rate for both train and test sets. Bottom: percentage of correct recalls (left), reasoning (center), and final answers (right).

## Task 14: "Among the list of *x*, is there *y*?"

**Category:** Listing

**Description:** The goal is to determine if a person ate a given fruit or not, among the list of fruits they ate.

| Template | Example |
|---|---|
| **Story:**
[Task 14] {*name*}'s Snacks
{*name*} ate {*fruit*} at {*time*}.
. . .
{*name*} ate {*fruit*} at {*time*}. | **Story:**
[Task 14] Tom's Snacks
Tom ate an apple at 8am.
Tom ate a pear at 10am.
Tom ate an orange at 2pm. |
| **Question:**
[Task 14] Among the snacks that {*name*} ate, is there a {*q_fruit*}? | **Question:**
[Task 14] Among the snacks that Tom ate, is there a banana? |
| **Answer:**
{*story*}
The answer is {*answer*}. | **Answer:**
Tom ate an apple at 8am.
Tom ate a pear at 10am.
Tom ate an orange at 2pm.
The answer is no. |

### Details

- {*time*} is sampled without replacement from ["8am", "10am", "12pm", "2pm"].
- {*fruit*} is a randomly sampled from ["an apple", "a pear", "an orange", "a banana", "a cherry"] for each statement.
- {*q_fruit*} is a randomly sampled from ["an apple", "a pear", "an orange", "a banana", "a cherry"].
- {*answer*} is a "yes" or "no".
- The story can have between 2 and 4 sentences, excluding the title. Sentences are ordered by {*time*}.

Table 16: Templates for generating Task 14 stories, questions, and answers, with an example provided.

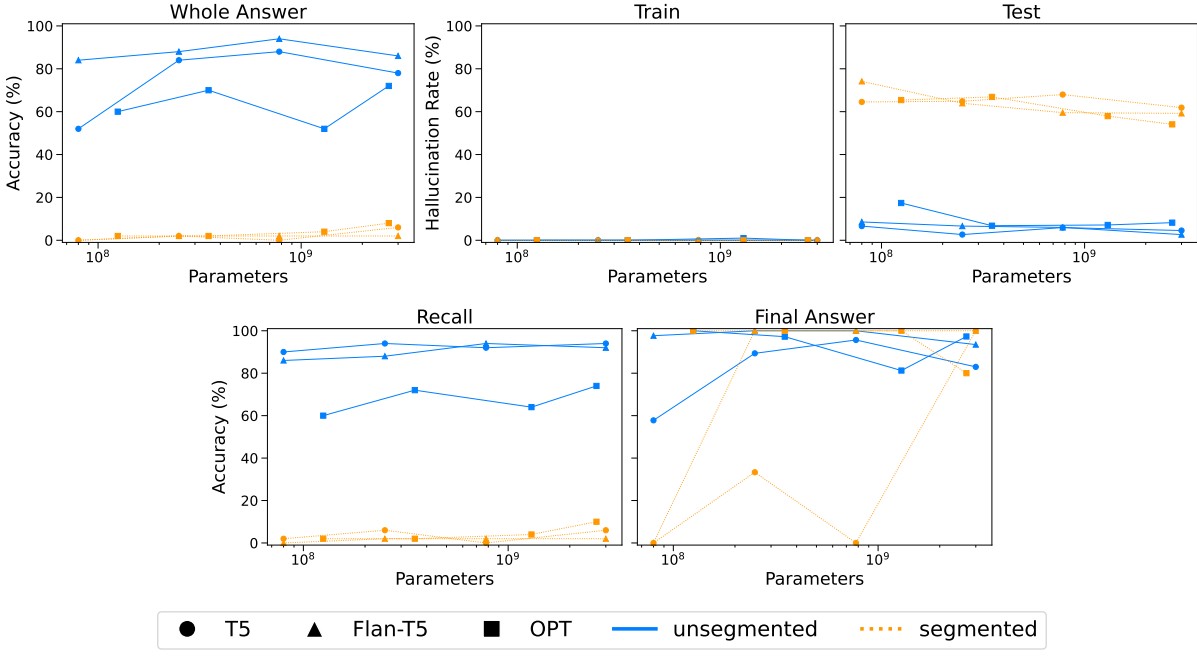

Figure 18: Task 14 results. Top left: percentage of correct answers. Top right: hallucination rate for both train and test sets. Bottom: percentage of correct recalls (left) and final answers (right).

**Task 15: "Among the list of *x*, is there only *y*?"**

**Category:** Uniqueness

**Description:** The goal of the task is to determine if the person only received a given grade in either their science or language courses.

| Template | Example |
|---|---|
| **Story:**
[Task 15] {*name*}'s Grades
{*name*} got {*grade*} in {*language_course*}.
. . .
{*name*} got {*grade*} in {*language_course*}.
{*name*} got {*grade*} in {*science_course*}.
. . .
{*name*} got {*grade*} in {*science_course*}. | **Story:**
[Task 15] Tom's Grades
Tom got an A in English.
Tom got an A in Spanish.
Tom got an B in Biology.
Tom got an A in Physics. |
| **Question:**
[Task 15] Did {*name*} only get {*q_grade*} in {*course_type*} courses? | **Question:**
[Task 15] Did Tom only get A in science courses? |
| **Answer:**
{*story*}
The answer is {*answer*}. | **Answer:**
Tom got an A in English.
Tom got an A in Spanish.
Tom got an B in Biology.
Tom got an A in Physics.
The answer is no. |

**Details**

- {*language_course*} is randomly sampled without replacement from ["English", "Spanish", "French"].
- {*science_course*} is randomly sampled without replacement from ["Biology", "Physics", "Chemistry"].
- {*grade*} is randomly chosen to be "A" or "B" for each statement.
- {*q_grade*} is randomly chosen to be "A" or "B" for each statement.
- {*course_type*} is randomly chosen to be "science" or "language".
- {*answer*} is either "yes" or "no".
- The story can contain 2 to 3 sentences about language courses and 2 to 3 sentences about science courses.

Table 17: Templates for generating Task 15 stories, questions, and answers, with an example provided.

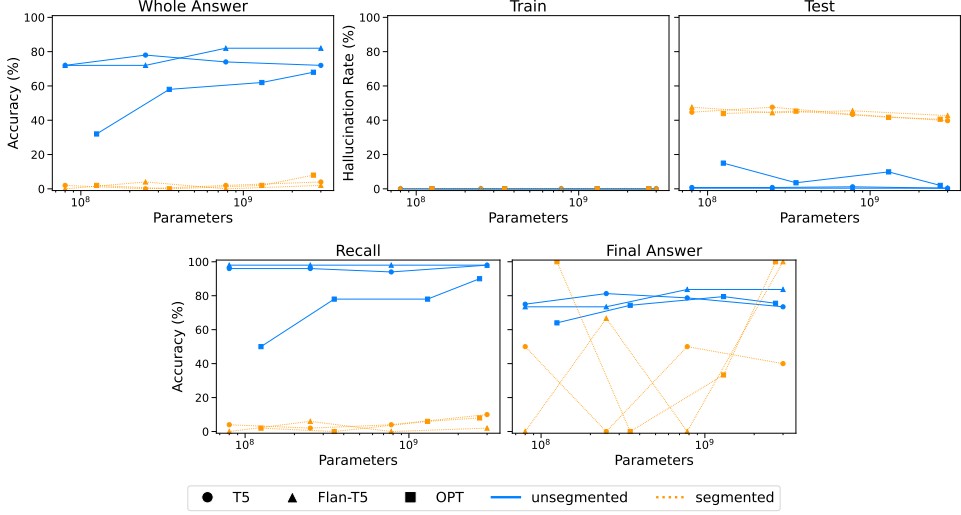

Figure 19: Task 15 results. Top left: percentage of correct answers. Top right: hallucination rate for both train and test sets. Bottom: percentage of correct recalls (left) and final answers (right).



**Task 16: "Is *x* the same as *y*?"**



**Category:** Ranking
**Description:** The goal is to determine if the person went as many times to the beach as to the cinema.

| Template | Example |
|---|---|
| **Story:**
[Task 16] {*name*}'s Activities
{*day*}, {*name*} went to {*place*}
…
{*day*}, {*name*} went to {*place*} | **Story:**
[Task 16] Tom's Activities
Monday, Tom went to the beach.
Tuesday, Tom went to the beach.
Wednesday, Tom went to the cinema.
Thursday, Tom went to the park.
Friday, Tom went to the cinema. |
| **Question:**
[Task 16] Did {*name*} go to the beach as many days as to the cinema? | **Question:**
[Task 16] Did Tom go to the beach as many days as to the cinema? |
| **Answer:**
{*story*}
The answer is {*answer*}. | **Answer:**
Monday, Tom went to the beach.
Tuesday, Tom went to the beach.
Wednesday, Tom went to the cinema.
Thursday, Tom went to the park.
Friday, Tom went to the cinema.
The answer is yes. |



**Details**



- {*day*} can be any day of the week.
- {*place*} is randomly sampled with replacement from ["cinema", "park", "beach"], but "cinema" and "beach" must each be sampled at least once and no more than three times.
- {*answer*} is either "yes" or "no".
- The story can contain 4 to 5 sentences, excluding the title. Sentences are ordered by {*day*}.

Table 18: Templates for generating Task 16 stories, questions, and answers, with an example provided.

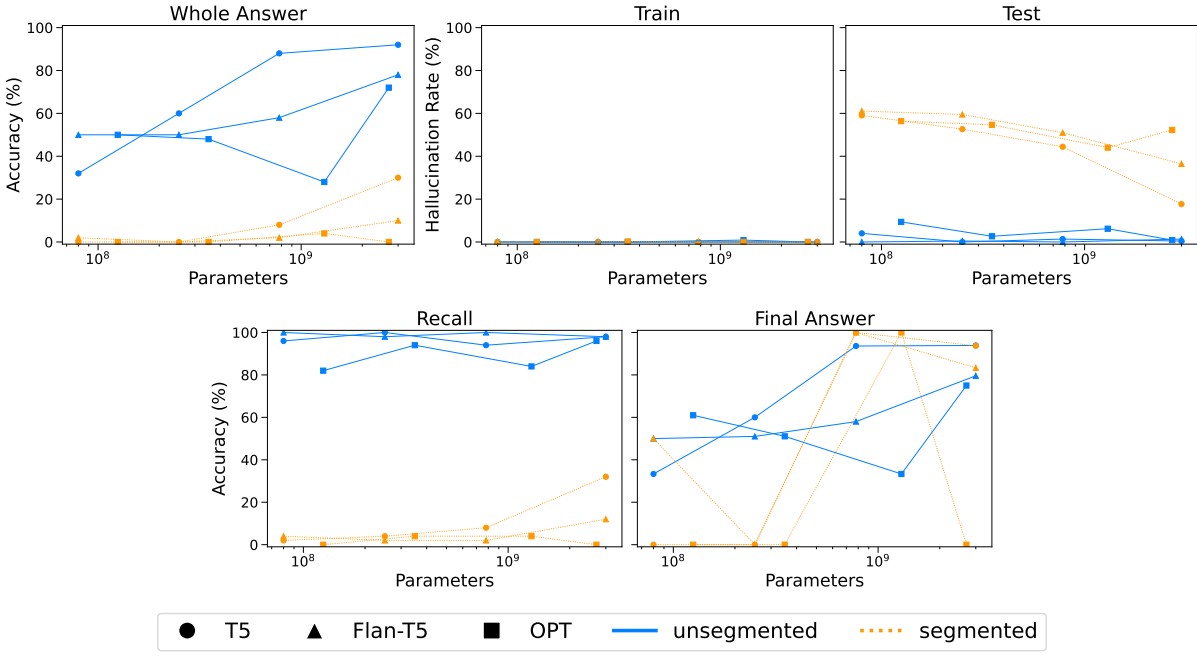

Figure 20: Task 16 results. Top left: percentage of correct answers. Top right: hallucination rate for both train and test sets. Bottom: percentage of correct recalls (left) and final answers (right).

**Task 17: "What is the state of *x* when *y* happens?"**

**Category:** Temporal

**Description:** The objective of this task is to identify what the individual was wearing at the moment the storm began.

| Template | Example |
|---|---|
| **Story:**
[Task 17] {*name*}'s Outfits
{*time*}, {*name*} is wearing {*clothing*}.
…
{*time*}, the storm starts.
…
{*time*}, {*name*} is wearing {*clothing*}. | **Story:**
[Task 17] Tom's Outfits
8am, Tom is wearing a pyjama.
10am, Tom is wearing workout clothes.
12pm, Tom is wearing a bathrobe.
2pm, the storm starts.
4pm, Tom is wearing a raincoat. |
| **Question:**
[Task 17] What was {*name*} wearing when the storm started? | **Question:**
[Task 17] What was Tom wearing when the storm started? |
| **Answer:**
{*story*}
The answer is {*answer*}. | **Answer:**
8am, Tom is wearing a pyjama.
10am, Tom is wearing workout clothes.
12pm, Tom is wearing a bathrobe.
2pm, the storm starts.
4pm, Tom is wearing a raincoat.
The answer is a bathrobe. |

**Details**

- {*time*} is randomly sampled without replacement from ["8am", "9am", "10am", "11am", "12pm", "1pm", "2pm", "3pm", "4pm", "5pm"].
- {*clothing*} is a randomly sampled without replacement from ["a pyjama", "workout clothes", "a bathrobe", "a raincoat"].
- The statement "{*time*}, the storm starts." is randomly positioned within the story but can appear anywhere from the first to the penultimate sentence.
- {*answer*} is the {*clothing*} mentioned in the sentence immediately preceding "{*time*}, the storm starts".
- The story will consist of 4 to 5 sentences, excluding the title, and sentences are sequenced according to {*time*}.

Table 19: Templates for generating Task 17 stories, questions, and answers, with an example provided.

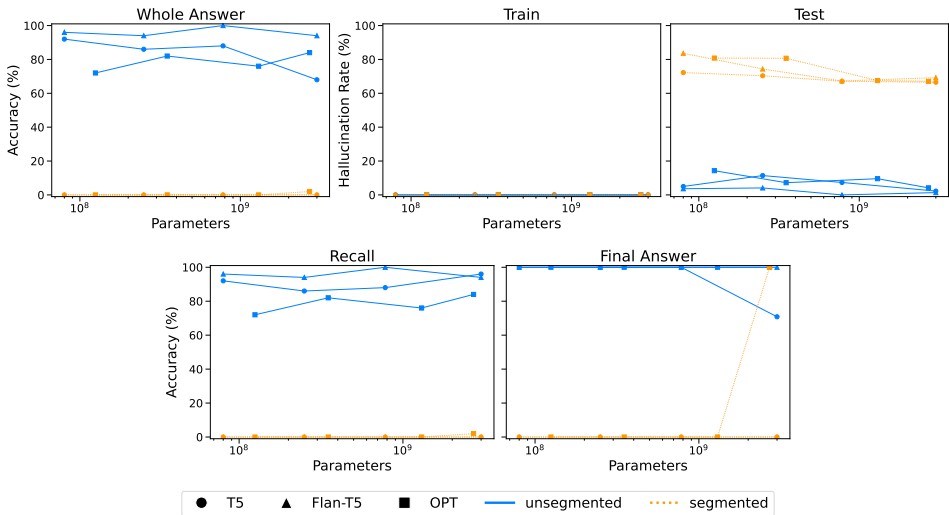

Figure 21: Task 17 results. Top left: percentage of correct answers. Top right: hallucination rate for both train and test sets. Bottom: percentage of correct recalls (left) and final answers (right).

## Task 18: "If *x* had/hadn't happened, would *y* have happened?"

**Category:** Causal

**Description:** This task requires determining whether, on a specific day, a person would have had a certain amount of money had they not sold a particular item.

| Template | Example |
|---|---|
| **Story:** | **Story:** |
| [Task 18] {*name*}'s Money | [Task 18] Tom's Money |
| {*day*}, {*name*} sold {*item*} for {*price*}$. | Monday, Tom sold a pencil for 2$. |
| ... | Tuesday, Tom sold an eraser for 1$. |
| {*day*}, {*name*} sold {*item*} for {*price*}$. | Wednesday, Tom sold a marker for 3$. |
| | Thursday, Tom sold a staple for 1$. |
| **Question:** | **Question:** |
| [Task 18] If {*name*} hadn't sold {*q_item*}, would they have {*q_money*}$ on {*q_day*}? | [Task 18] If Tom hadn't sold a staple, would they have 6$ on Wednesday? |
| **Answer:** | **Answer:** |
| {*story*} | Monday, Tom sold a pencil for 2$. |
| {*reasoning*} | Tuesday, Tom sold an eraser for 1$. |
| The answer is {*answer*}. | Wednesday, Tom sold a marker for 3$. |
| | Thursday, Tom sold a staple for 1$. |
| | 2 + 1 + 3 = 6. |
| | The answer is yes. |

### Details

- {*day*} is randomly sampled without replacement from ["Monday", "Tuesday", "Wednesday", "Thursday"].
- {*item*} is randomly sampled without replacement from ["a pencil", "an eraser", "a marker", "a staple"].
- {*price*} is a randomly sampled integer value between 1 and 3.
- {*q_item*} is randomly sampled from ["a pencil", "an eraser", "a marker", "a staple"].
- {*q_money*} is some integer value between 3 and 8.
- {*q_day*} is randomly selected among the sampled {*day*}.
- {*reasoning*} is the summation of the {*price*} up to (and including) {*q_day*}, but excludes the {*price*} corresponding to {*q_item*}.
- {*answer*} is either "yes" or "no".
- The story consists of 3 to 4 sentences, not counting the title, and sentences are listed chronologically by {*day*}.

Table 20: Templates for generating Task 18 stories, questions, and answers, with an example provided.

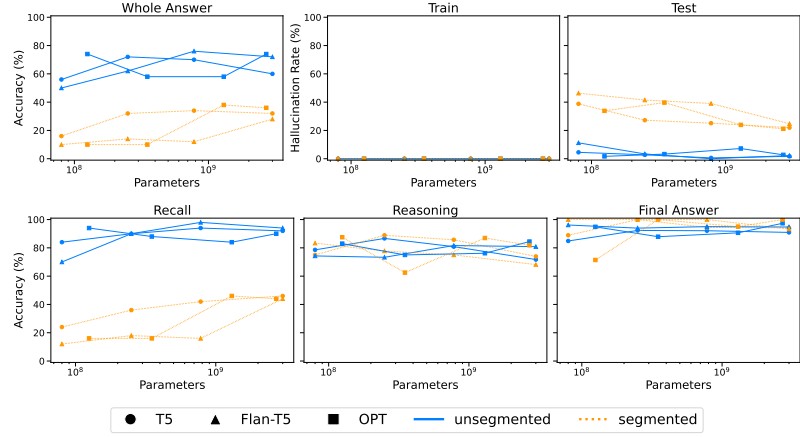

Figure 22: Task 18 results. Top left: percentage of correct answers. Top right: hallucination rate for both train and test sets. Bottom: percentage of correct recalls (left), reasoning (center), and final answers (right).

| Question | Target Answer | Model Answer |
|---|---|---|
| [Task 1] Which days did Hailey work from home? | Hailey worked from home on Wednesday. The answer is Wednesday. | Hailey worked from home on Wednesday. **Hailey worked from home on Friday.** The answer is Wednesday and Friday. |
| [Task 2] How many times did Kozma go fishing? | Kozma went fishing on Monday. Kozma went hiking on Tuesday. Kozma went fishing on Wednesday. Kozma went fishing on Friday. Kozma went hiking on Saturday. The answer is 3. | Kozma went fishing on Monday. Kozma went hiking on Tuesday. Kozma went fishing on Wednesday. **Kozma went hiking on Thursday.** Kozma went fishing on Friday. The answer is 2. |
| [Task 3] Does Artemia have more meetings with co-worker A or B? | 1:00 PM - Artemia fills up some forms. 2:00 PM - Artemia has a meeting with co-worker A. 3:00 PM - Artemia has a meeting with co-worker A. 4:00 PM - Artemia has a meeting with co-worker B. 5:00 PM - Artemia fills up some forms. The answer is A. | 1:00 PM - Artemia fills up some forms. 2:00 PM - Artemia has a meeting with co-worker A. 3:00 PM - **Artemia fills up some forms**. 4:00 PM - Artemia has a meeting with co-worker B. The answer is neither. |
| [Task 4] Does Wahoo buy a house after they go on a vacation? | Wahoo goes on a vacation in January. Wahoo buys a house in March. March is after January. The answer is yes. | Wahoo goes on a vacation in January. Wahoo buys a house in March. **Wahoo gets married in October.** March is after January. The answer is yes. |
| [Task 5] When Mayr is in Vancouver, is Grant in Rome? | Mayr was in New York on Monday. Mayr was in Vancouver on Tuesday. Mayr was in Paris on Wednesday. Grant was in Rome on Monday. Grant was in Tokyo on Tuesday. Grant was in Los Angeles on Wednesday. Those are different days. The answer is no. | Mayr was in **Vancouver** on Monday. Mayr was in **New York** on Tuesday. Mayr was in Paris on Wednesday. Grant was in Rome on Monday. Grant was in **Los Angeles** on Tuesday. Those are the same days. The answer is yes. |
| [Task 6] Adlar visits a friend on Friday. Is it the only time that Adlar visits a friend? | Adlar goes to the park on Monday. Adlar visits a friend on Tuesday. Adlar plays golf on Wednesday. Adlar goes to the park on Thursday. Adlar visits a friend on Friday. The answer is no. | Adlar goes to the park on Monday. Adlar visits a friend on Tuesday. Adlar **visits a friend** on Wednesday. Adlar **plays golf** on Thursday. Adlar **goes to the park** on Friday. The answer is no. |
| [Task 7] Between going for a walk and making tea, does Braili read a book? | Morning, Braili goes for a walk. Afternoon, Braili reads a book. Evening, Braili makes tea. The answer is yes. | Morning, Braili goes for a walk. **Noon, Braili makes a phone call.** Afternoon, Braili **makes tea**. Evening, Braili **reads a book**. The answer is yes. |
| [Task 8] How much time passes between Damiyon made a phone call and sent an email? | At 1pm, Damiyon sent an email. At 3pm, Damiyon started a video chat. At 4pm, Damiyon wrote a letter. At 5pm, Damiyon made a phone call. 5 - 1 = 4. The answer is 4. | At 1pm, Damiyon sent an email. At **2**pm, Damiyon **made a phone call**. At 4pm, Damiyon wrote a letter. At 5pm, Damiyon **started a video chat**. 2 - 1 = 1. The answer is 1. |
| [Task 9] At what time does Tia ask for the bill? | Tia arrived at the restaurant at 6:08 PM. 3 minutes after arriving, Tia ordered a drink. 2 minutes after ordering a drink, Tia ordered a sandwich. 3 minutes after ordering a sandwich, Tia asked for the bill. 3 + 2 + 3 = 8. The answer is 6:16 PM. | Tia arrived at the restaurant at 6:08 PM. **2** minutes after arriving, Tia ordered a drink. **3** minutes after ordering a drink, Tia **asked for the bill**. 2 + 3 = 5. The answer is 6:13 PM. |

Table 21: Hallucination examples from models trained on segmented stories for Tasks 1 to 9. The Target Answer is provided for comparison, with hallucinations highlighted in bold. For hallucination examples corresponding to Tasks 10 to 18, see Table 22.

| Question | Target Answer | Model Answer |
|---|---|---|
| [Task 10] The second time that Demontre went canoeing, who else was there? | Monday, Demontre went canoeing with Emelita. Tuesday, Demontre went hunting with Taifa. Thursday, Demontre went canoeing with Maibelle. Friday, Demontre went hunting with Ebere. Sunday, Demontre went hunting with Amyty. The answer is Maibelle. | Tuesday, Demontre went **canoeing** with **Maibelle**. Thursday, Demontre went canoeing with **Xian**. Friday, Demontre went hunting with **Neria**. Sunday, Demontre went hunting with **Sidoney**. The answer is Xian. |
| [Task 11] Every time Lamya drives to pharmacy, is it always in a minivan? | Tuesday, Lamya drives to pharmacy in a minivan. Wednesday, Lamya drives to pharmacy in a SUV. Thursday, Lamya drives to pharmacy in a minivan. The answer is no. | **Monday, Lamya drives to pharmacy in a SUV.** Tuesday, Lamya drives to pharmacy in a minivan. Wednesday, Lamya drives to **grocery store** in a SUV. Thursday, Lamya drives to pharmacy in a **SUV**. The answer is no. |
| [Task 12] After how many days does Rooster join Osmar? | Monday, Osmar is alone. Tuesday, Osmar is alone. Wednesday, Osmar is alone. The answer is never. | Monday, Osmar is alone. Tuesday, Osmar is alone. Wednesday, Osmar is alone. **Thursday, Rooster arrives.** The answer is 4. |
| [Task 13] Is Person D the third person Marayna meets? | Marayna meets Person B in the morning. Marayna meets Person C at noon. Marayna meets Person D in the afternoon. Person D is the third. The answer is yes. | Marayna meets Person B in the morning. Marayna meets Person **A** at noon. Marayna meets Person **C** in the afternoon. Person D is the first. The answer is no. |
| [Task 14] Among the snacks that Kornelis ate, is there an orange? | Kornelis ate a banana at 8am. Kornelis ate an apple at 2pm. The answer is no. | Kornelis ate a banana at 8am. Kornelis ate an apple at **10am**. **Kornelis ate a cherry at 12pm.** The answer is no. |
| [Task 15] Did Corrine only get B in language courses? | Corrine got an A in English. Corrine got a B in Spanish. Corrine got an A in French. Corrine got an A in Biology. Corrine got an A in Physics. Corrine got an A in Chemistry. The answer is no. | Corrine got an A in English. Corrine got a B in Spanish. Corrine got an A in French. Corrine got an A in Biology. Corrine got **a B** in Physics. The answer is no. |
| [Task 16] Did Trella go to the beach as many days as to the cinema? | Monday, Trella went to the beach. Tuesday, Trella went to the cinema. Wednesday, Trella went to the beach. Thursday, Trella went to the park. Friday, Trella went to the cinema. The answer is yes. | Monday, Trella went to the beach. Tuesday, Trella went to the cinema. Wednesday, Trella went to the **park**. Thursday, Trella went to the park. Friday, Trella went to the **park**. The answer is no. |
| [Task 17] What was Kyra wearing when the storm started? | 9am, Kyra is wearing a raincoat. 12pm, the storm starts. 2pm, Kyra is wearing workout clothes. 3pm, Kyra is wearing a bathrobe. The answer is a raincoat. | 9am, Kyra is wearing a raincoat. **10am, Kyra is wearing a pyjama.** 12pm, the storm starts. 2pm, Kyra is wearing workout clothes. The answer is a pyjama. |
| [Task 18] If Rheanna hadn't sold a a staple, would they have 8$ on Wednesday? | Monday, Rheanna sold an eraser for 3$. Tuesday, Rheanna sold a marker for 3$. Wednesday, Rheanna sold a staple for 2$. 3 + 3 = 6. The answer is no. | Monday, Rheanna sold an eraser for 3$. Tuesday, Rheanna sold a marker for **2**$. Wednesday, Rheanna sold a staple for **1**$. 3 + 2 = 5. The answer is no. |

Table 22: Hallucination examples from models trained on segmented stories for Tasks 10 to 18. The Target Answer is provided for comparison, with hallucinations highlighted in bold. For hallucination examples corresponding to Tasks 1 to 9, see Table 21.

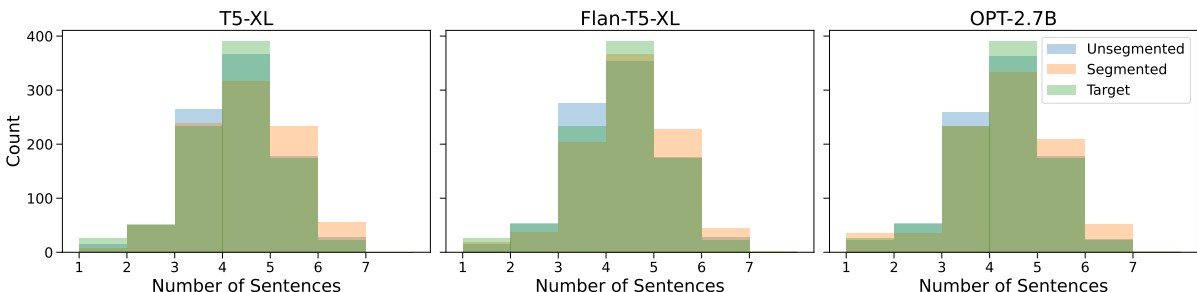

Figure 23: Comparison of the number of sentences in the recall part of answers from three models: T5-XL (left), Flan-T5-XL (center), and OPT-2.7B (right). This compares the target distribution with models trained on unsegmented and segmented stories. Similar patterns were observed for other model sizes. There is no significant difference between these distributions, suggesting that training on segmented stories does not affect recall length.