# OpenReview forum: "EpiK-Eval: Evaluation for Language Models as Epistemic Models"
_EMNLP/2023/Conference — EMNLP 2023 Main_

### Official Review · Reviewer_tdge · 2023-08-02

**Typos Grammar Style And Presentation Improvements:** •	The two colors in Figure 2 are diff…
**Soundness:** 3

**Excitement:**

3: Ambivalent: It has merits (e.g., it reports state-of-the-art results, the idea is nice), but there are key weaknesses (e.g., it describes incremental work), and it can significantly benefit from another round of revision. However, I won't object to accepting it if my co-reviewers champion it.

**Paper Topic And Main Contributions:**

The authors investigate the ability of LMs to learn from fragmented information, in other words, to identify information belonging together and combining for consistent inference. For this investigation a corpus of stories is fragmented into sentences and LMs (T5 and Flan-T5) trained on the fragments and compared to the same models trained on the un-fragmented stories. The models are evaluated on QA tasks that requires the combination of multiple sentences in a story.
The authors find that LMs perform poorly on the QA tasks when learning from fragmented data. They conclude that the future work should investigate adapting the training objective of LMs to learn context beyond just one input document.

**Questions For The Authors:**

•	In Figure 3, the performance on the different QA types are compared. We can see a difference in performance for the non-baseline models on the different types. Can this difference be interpreted in any way?
•	How are the stories produced?

**Reasons To Accept:**

The ability to combine information seen in different documents during training is crucial for many applications. This is the first study that investigates LMs’ capability to perform this information combination effectively.

**Reasons To Reject:**

The corpus used for this study, EpiK-Eval, is extremely fragmented – the stories are split into their sentences. Even for a human it would be difficult to identity which of these fragments belong to the same story and need to be considered together to solve the QA task. A more realistic scenario with better insights would be multiple fragmentation ratios, with sentence splitting being one extreme and no splitting being the other. For example, if stories are split into two, the model may be able to identify that the two fragments need to be combined during inference.

EDIT: The authors clarified that each fragment contains the story title, meaning that for a human it would indeed be an easy task to identify fragments belonging together.

**Reproducibility:**

4: Could mostly reproduce the results, but there may be some variation because of sample variance or minor variations in their interpretation of the protocol or method.

**Reviewer Confidence:**

4: Quite sure. I tried to check the important points carefully. It's unlikely, though conceivable, that I missed something that should affect my ratings.

---

> ### Author Rebuttal · Authors · 2023-08-29
>
> We would like to extend our sincere appreciation to the reviewers for their thorough and invaluable feedback. It brings us great joy to note that our work has captured the interest of the reviewers (R1 and R2), been recognized for its novelty (R1 and R3), and acknowledged as insightful (R1 and R2). Furthermore, we are delighted to learn that the matter of addressing knowledge consolidation in LLMs has been recognized as "an important topic that needs to be address" (R2). We have meticulously examined all the concerns raised by the reviewers and have diligently endeavored to address these issues, as detailed in the discussion below.
>
> **Reviewer’s Comment: “A more realistic scenario with better insights would be multiple fragmentation ratios, with sentence splitting being one extreme and no splitting being the other. For example, if stories are split into two, the model may be able to identify that the two fragments need to be combined during inference.”**
>
> **Response:** This is a very interesting point which we had not thought about. An approximation of this setup is to group performance by the number of sentences in the stories. We ran this setup and obtained the following results:
>
> #### Flan-T5 Accuracy (Unsegmented Stories)
> |           |    Small |   Base  |  Large  |  XL     |
> |---|---|---|---|---|
> | 1 sentences |   77.97% |  89.83% |  96.61% | 94.07%  |
> | 2 sentences |  100.00% | 100.00% | 100.00% | 93.33%  |
> | 3 sentences |   85.47% |  92.31% |  92.31% | 96.58%  |
> | 4 sentences |   84.00% |  92.00% |  96.00% | 96.00%  |
> | 5 sentences |   59.78% |  60.00% |  81.78% | 81.78%  |
> | 6 sentences |   62.00% |  70.00% |  69.00% | 78.00%  |
> | Total       |   69.11% |  73.44% |  85.56% | 86.67%  |
>
> #### Flan-T5 Accuracy (Segmented Stories)
> |           |    Small |   Base  |  Large  |  XL     |
> |---|---|---|---|---|
> | 1 sentences | 14.41% | 26.27% | 22.88% | 32.20% |
> | 2 sentences | 33.33% | 26.67% | 26.67% | 26.67% |
> | 3 sentences | 34.19% | 37.61% | 41.88% | 50.43% |
> | 4 sentences | 11.00% | 22.00% | 29.00% | 34.00% |
> | 5 sentences | 10.22% |  9.56% | 13.11% |  9.33% |
> | 6 sentences | 4.00% |  2.00% |  1.00% |  6.00% |
> | Total       | 13.67% | 16.22% | 18.78% | 20.33% |
>
> #### Flan-T5 Hallucination Rate (Unsegmented Stories)
> |           |    Small |   Base  |  Large  |  XL     |
> |---|---|---|---|---|
> | 1 sentences | 25.58% | 7.14% | 3.49% | 4.82%
> | 2 sentences |  0.00% | 0.00% | 0.00% | 0.00%
> | 3 sentences | 6.00% | 2.85% | 2.56% | 1.14%
> | 4 sentences | 10.75% | 7.00% | 3.00% | 2.50%
> | 5 sentences | 9.44% | 6.80% | 4.61% | 4.03%
> | 6 sentences | 6.19% | 1.37% | 0.69% | 0.00%
> | Total       | 9.04% | 5.54% | 3.56% | 2.94%
>
> #### Flan-T5 Hallucination Rate (Segmented Stories)
> |           |    Small |   Base  |  Large  |  XL     |
> |---|---|---|---|---|
> | 1 sentences | 72.55% | 72.55% | 64.36% | 55.88%
> | 2 sentences | 27.78% | 27.78% | 27.03% | 26.32%
> | 3 sentences | 22.25% | 22.25% | 19.54% | 15.32%
> | 4 sentences | 59.00% | 59.00% | 43.00% | 36.25%
> | 5 sentences | 62.48% | 62.48% | 57.19% | 57.44%
> | 6 sentences | 41.24% | 41.24% | 41.75% | 37.97%
> | Total       | 54.97% | 54.97% | 49.61% | 47.85%
>
> where Accuracy is as in our paper (Section 3): the percentage of correct answers, where a correct answer means that it exactly matches the target answer. Hallucination Rate is also as in the paper (Section 3): the rate of sentences in the model's answer that weren't in the target answer. We note that the accuracy does seem to lower in the 5-6 sentence range. This makes sense given that the model's answer must exactly match the target answer. Since the hallucination rate is not zero, as more sentences need to be recalled, the likelihood of making a mistake increases. As for the hallucination rate, interestingly it fluctuates. The rate starts relatively higher for 1 sentence, then lowers until 3 sentences, comes back up until 5 sentences and lowers a bit for 6 sentences. Since other variables can affect the performance though, for example the number of facts to recall in a given sentence might vary between different tasks, we cannot make strong conclusions between the hallucination rate and the number of sentences in the story. The ideal experiment will be to do what the reviewer has proposed, which is given a story, measure if the performance changes depending on how many times it is segmented. We plan on running this experiment and adding it to the paper.
>
> **Reviewer’s Comment: “In Figure 3, the performance on the different QA types are compared. We can see a difference in performance for the non-baseline models on the different types. Can this difference be interpreted in any way?”**
>
> **Response:** The goal of Figure 3 was twofold. First, we wanted to compare the difficulty of the different tasks. For example, some types of relationships such as temporal or causal might be more challenging for the models. The problem with Figure 3 right now is that we only look at whether or not the entire answer matches the target answer. So if some tasks have longer stories on average than others, those can be more difficult with respect to hallucinations because more sentences need to be recalled. Thus, in order to study solely the reasoning aspect of these tasks, we would need to only consider cases where the model has successfully recalled the story and then look at the reasoning performance. The models trained on the unsegmented stories are a good approximation of this, since they have a very high success rate at recalling the story without hallucinating any part. If we thus look at the performance of these models, we can guess that some categories seem more difficult than others specifically because the reasoning is more challenging. But instead of guessing, we can simply measure the reasoning performance of these models (number of reasoning and final answers that match the target) excluding answers with an issue in the recall part of the answer. Doing so, we get the following results:
> #### Flan-T5 Reasoning Accuracy per Category (Unsegmented Stories)
> |           |    Small |   Base  |  Large  |  XL     |
> |---|---|---|---|---|
> Retrieval | 91.15% | 100.00% | 100.00% | 100.00%
> Comparison & Ranking | 80.71% |   0.00% |  62.50% |  90.00%
> Temporal | 82.53% |  96.00% |  95.83% |  90.00%
> Parallel | 100.00% | 100.00% | 100.00% | 100.00%
> Causal | 100.00% | 100.00% | 100.00% | 100.00%
> Exclusivity | 65.25% | 100.00% | 100.00% | 100.00%
> Detail Recognition | 60.61% | -       | -       | -
> Consistency | 100.00% | 100.00% | 100.00% | 100.00%
> Interrelation | 100.00% | 100.00% | 100.00% | 100.00%
>
> As we can see, the larger models get very good reasoning accuracy in all categories, except comparison & ranking which seems a bit more challenging, but the XL model eventually performs well on it. A dash in the table means that the models failed to recall the story properly and thus the reasoning part is ignored since we only want to evaluate the reasoning when the model has recalled the correct information first. We will add this analysis to the paper.
>
> The second goal of Figure 3 was to verify if for some categories/tasks, the models trained on segmented stories performed more closely to the models trained on unsegmented stories. This is true for certain categories. The reason for a difference in gap could be: because the stories are longer or contain more facts and thus it is more difficult for the model to recall them without making a mistake or because the reasoning is more challenging. If we look at the average number of sentences of stories per category, we get:
> | Category | Sentences on average |
> |---|---|
> | Retrieval |            3.33 |
> | Comparison & Ranking | 5.50 |
> | Temporal |             4.75 |
> | Parallel |             5.00 |
> | Causal |               5.00 |
> | Exclusivity |          5.5 |
> | Detail recognition |   5.00 |
> | Consistency |          3.00 |
> | Interrelation |        4.00 |
>
> which does not seem to correlate with the gap between the models trained on segmented and unsegmented stories. As we saw earlier, the reasoning aspect also does not seem to be a challenge for the larger models. Finally, we could look at the number of facts to recall per story to see if this correlates with the gap in performance:
> #### Flan-T5 Hallucination Rate (Unsegmented Stories)
> |           |    Small |   Base  |  Large  |  XL     |
> |---|---|---|---|---|
> Retrieval | 11.04% |  3.12% |  1.00% | 1.52%
> Comparison & Ranking | 6.07% |  5.71% |  0.78% | 2.34%
> Temporal | 5.76% |  2.72% |  3.36% | 6.05%
> Parallel | 7.20% |  0.00% |  0.00% | 0.00%
> Causal | 12.80% | 10.80% |  3.60% | 0.80%
> Exclusivity | 7.30% |  1.57% |  3.14% | 2.40%
> Detail Recognition | 20.40% | 24.40% | 14.00% | 0.40%
> Consistency | 12.00% |  6.00% |  5.33% | 2.67%
> Interrelation | 21.50% | 14.00% |  6.00% | 5.00%
>
> #### Flan-T5 Reasoning Accuracy per Category (Segmented Stories)
> |           |    Small |   Base  |  Large  |  XL     |
> |---|---|---|---|---|
> Retrieval | 50.16% | 42.80% | 45.46% | 39.59%
> Comparison & Ranking | 52.20% | 51.92% | 44.79% | 36.41%
> Temporal | 63.38% | 58.66% | 59.38% | 59.19%
> Parallel | 43.60% | 50.80% | 46.00% | 60.00%
> Causal | 67.20% | 73.60% | 56.00% | 54.80%
> Exclusivity | 51.94% | 56.74% | 51.00% | 52.00%
> Detail Recognition | 87.60% | 94.40% | 88.40% | 85.20%
> Consistency | 22.67% | 20.00% | 14.67% |  8.67%
> Interrelation | 62.00% | 52.00% | 31.00% | 20.00%
>
> The hallucination rate does vary depending on the category. If we look at the consistency and interrelation results, the XL models trained on segmented stories hallucinate at a lower rate than in other categories and these same models perform relatively well compared to other categories in Figure 3. As for other categories, the pattern is less clear. To better understand the reason for the gap between models trained on segmented and unsegmented stories with respect to categories of relation, a per task analysis would be required as too many variables come into play. We believe though that this falls outside the scope of the paper, as the goal is to study the impact of segmentation on the performance of these models. The choice of task is somewhat irrelevant and the reasoning aspect certainly is. We will add these analyses and discussion to the paper.
>
> **Reviewer’s Comment: “The two colors in Figure 2 are difficult to distinguish.”**
>
> **Response:** We have switched the colors in Figure 2 from blue and light-blue to blue and orange.
>
> **Reviewer’s Comment: “How are the stories produced?”**
>
> **Response:** We have clarified this process in the updated section 3, which we have provided in the following response. Please refer to it. If any details are still unclear, the reviewer is more than welcome to let us know.
>
> **Reviewer’s Comment: “The corpus used for this study, EpiK-Eval, is extremely fragmented – the stories are split into their sentences. Even for a human it would be difficult to identity which of these fragments belong to the same story and need to be considered together to solve the QA task.”**
>
> **Response:** We have failed to acknowledge the following detail and apologize for it. Each story and story segment begins with the title of the story, such that the model can easily tell which story a given segment belongs to. For example, an unsegmented story will look like:
> > [Task 1] Tom's Afternoon
> >
> > 1 PM: Tom reads a book.
> >
> > 3 PM: Tom goes to the store.
> >
> > 5 PM: Tom is at the restaurant.
>
> And the story segments:
> > [Task 1] Tom's Afternoon, Part 1/3
> >
> > 1 PM: Tom reads a book.
>
> > [Task 1] Tom's Afternoon, Part 2/3
> >
> > 3 PM: Tom goes to the store.
>
> > [Task 1] Tom's Afternoon, Part 3/3
> >
> > 5 PM: Tom is at the restaurant.
>
> The character's name in each story is also unique and appears in each sentence, thus the model can always relate the story parts to the character's name. We’ve updated section 3 with this clarification as well as clarifying the entire data generation and evaluation process. See our updated section 3 below.
>
> ## 3 EpiK-Eval
> The EpiK-Eval benchmark presents a suite of novel, narrative-based diagnostic tasks, meticulously designed to evaluate a LM's capacity to construct a comprehensive, unified knowledge state.
>
> **Dataset:** Our benchmark comprises 18 tasks, which are questions about relations between facts and events in stories, for example: "Does x happen before/after y?". See Table 2 for the full list of tasks. For each of these, we generate 100 stories following a per task template. For example, Task 2 has the following template:
> > [Task 2] {name}'s Vacation
> >
> > {name} went {activity} on {day}.
> >
> > .
> >
> > .
> >
> > .
>
> where the first line is the story's title, the name is randomly sampled such that it is unique to each story and the activity and day in a sentence are randomly sampled from the list ['fishing', 'hiking'] and ['Monday', 'Tuesday', 'Wednesday', 'Thursday', 'Friday', 'Saturday', 'Sunday'] respectively. The story can have a random number of sentences, the range is pre-determined per task, for example, Task 2 stories can have between 3 and 5 sentences. Here is an example of a generated story for Task 2:
> > [Task 2] Tom's Vacation
> >
> > Tom went fishing on Monday.
> >
> > Tom went hiking on Wednesday.
> >
> > Tom went fishing on Saturday.
>
> Thus, with a 100 stories generated for each 18 tasks, we get 1800 stories, which we refer to as our dataset of unsegmented stories D_U = {x_1, x_2, ..., x_1800}. After having generated these stories, we also generate a second dataset which consists of the segmented version of these stories. For each given story, we segment it into its individual sentences and add a part number to the title. For example, given the previous story about Tom, we would get the following three text sequences:
> > [Task 2] Tom's Vacation, Part 1/3
> >
> > Tom went fishing on Monday.
>
> > [Task 2] Tom's Vacation, Part 2/3
> >
> > Tom went hiking on Wednesday.
>
> > [Task 2] Tom's Vacation, Part 3/3
> >
> > Tom went fishing on Saturday.
>
> We do this for all 1800 stories and get 6800 story segments, which form our dataset of story segments D_S = {s_1, s_2, ..., s_6800}.
>
> For each story, we also generate one question-answer pair. The questions are rephrasings of the task. For example, for Task 2 "How many times does x happen?", we have "How many times did {name} go fishing?". The question-answer pairs are also generated following a template. The template always consists of a question followed by the answer which itself has three parts: recall of the entire story, an optional reasoning part depending on the task and the final answer. For example, the template for question-answers in Task 2 is the following:
> > [Task 2] How many times did {name} go fishing?
> >
> > {story} The answer is {count}.
>
> and here is an example of a generated question-answer pair:
> > [Task 2] How many times did Tom go fishing?
> >
> > Tom went fishing on Monday. Tom went hiking on Wednesday. Tom went fishing on Saturday. The answer is 2.
>
> We also include in the Appendix a full list of templates as well as the values each placeholder variable (e.g., activities, days) can take, in order to generate stories and question-answer pairs for each task. Having generated one question per story, we have a total of 1800 question-answer pairs, which we randomly split into two sets: the validation and the test set. In order for the models to learn the answer format that they are expected to respect while answering the questions, we add question-answer examples to the training set. We thus generate an additional 1800 stories and question-answer pairs. We discard the stories and add these 1800 question-answer pairs to the training set, such that there are no overlaps between questions in the training, validation and test set.
>
> **Evaluation Process:** We evaluate pretrained LMs for their ability to consolidate knowledge. To do so, given a pretrained language model, we make two copies of it: M_U and M_S. We fine-tune M_U on the unsegmented stories and M_S on the segmented stories. The prior setting allows for all the information necessary to answer a given question to be found in a single text sequence without requiring the model to learn dependencies across multiple text sequences, while the latter setting requires consolidating the information from the narrative segments. Having both allows to measure the effect of information being spread across separate text sequences and the LMs' ability to consolidate this knowledge during inference, by measuring the gap in performance between both models.
>
> Both models are fine-tuned on their respective dataset, D_U and D_S, as well as the training set of question-answer examples. Thus, one epoch for M_U consists of 3600 samples (1800 stories + 1800 q/a examples) and one epoch for M_S of 8600 samples (6800 segments + 1800 q/a examples). Samples are shuffled such that a batch may contain a mix of stories and question-answer examples in the case of M_U or story segments and question-answer examples in the case of M_S. Models are fine-tuned with their respective pretraining objective. Specifically, in the case of encoder-decoder style models, the story's title (first line in the text sequence) is fed to the encoder and the decoder is expected to output the rest of the story in the case of M_U or the story segment in the case of M_S. As for question-answer pairs, the question is fed to the encoder and the model is expected to output the answer. For causal language models, they are simply expected to predict the next token in the given sequence, as is standard procedure. Precisely, for M_U, a text sequence is either an entire story or a question concatenated with its answer, while for M_S, a text sequence is either a story segment or a question concatenated with its answer.
>
> During fine-tuning, both models are also periodically evaluated on the validation set. Models are run in inference mode as described in the papers they were introduced in. We prompt the models with the questions from the validation set and their answers are compared to the target answers. For an answer to be deemed as correct, it must match the exact target answer. This is to capture potential recall and reasoning errors as well as verifying the final answer. It is important for us to evaluate M_S's ability to consolidate the separate story segments, which is why we require the model to recall the entire story when answering a question. Here, M_U serves as an upper-bound on the performance and any potential gap in performance between it and M_S showcases the added difficulty of consolidating knowledge from the story segments. The number of correctly answered questions over the total number of questions is referred to as the accuracy. We also measure an additional metric, which we refer to as the hallucination rate. Given an answer, consider only the recall part of the answer and disregard the reasoning part and the final answer. Our hallucination rate is the number of recalled sentences that contains an error (does not match with the actual sentence in the narrative) over the total number of recalled sentences. This metric provides a more fine-grained examination of the recall and knowledge consolidation capabilities of the model. We want to evaluate if the model is more likely to hallucinate a fact, event or segments when recalling these from multiple training sequences (segmented setup) versus a single training sequence (unsegmented setup).
>
> Once both models have been fine-tuned, we take the best performing checkpoint of each model on the validation set and evaluate these on the test set. This is done in the same manner as the validation, except that the questions are from the test set.

---

### Official Review · Reviewer_1wi7 · 2023-08-03

**Soundness:** 3

**Excitement:**

4: Strong: This paper deepens the understanding of some phenomenon or lowers the barriers to an existing research direction.

**Paper Topic And Main Contributions:**

This paper proposes a benchmark called Epik-Eval that aims at measuring the ability of LLMs to consolidate knowledge and answer complex questions, which require to build a "knowledge state". The authors also propose two baselines based on T5, one of them trained on the non-segmented version of the dataset and the other one on the segmented train set.

Experimentation shows how the segmented version of the LLM incurs in lower perfromance and hallucinates more. This behavior is amelliorated by the instruct-following versions of the model, but this improvement allegedly comes from a better understanding of the question rather thatn from an improved ability to aggregate knowledge. The paper concludes with a number of recommendations, which include adding this kind of tasks involving knowledge consolidation as a predictive goal of LLM pretraining.

I wonder how the dataset and the task of knowledge consolidation relates to multi-hop question answering and datasets like HotpotQA. While Epik-Eval makes emphasis on a sequence of events with knowledge that needs to be consolidated in order to reach a final output, HotpotQA requires the model to extract information at each step of the reasoning chain that allows addressing such steps in order to answer the overall multi-hop question.

**Reasons To Accept:**

+ Knowledge consolidation in LLMs is an important topic that needs to be address
+ Evaluation indicates the benchmark is actually useful to hightlight LLMs' ability to deal with this task
+ Interesting final recommendations

**Reasons To Reject:**

- I would have liked to see more LLMs evaluated on the dataset in this paper
- Related work should probably look at work like multi-hop QA and existing datasets

**Reproducibility:**

4: Could mostly reproduce the results, but there may be some variation because of sample variance or minor variations in their interpretation of the protocol or method.

**Reviewer Confidence:**

4: Quite sure. I tried to check the important points carefully. It's unlikely, though conceivable, that I missed something that should affect my ratings.

---

> ### Author Rebuttal · Authors · 2023-08-29
>
> We would like to extend our sincere appreciation to the reviewers for their thorough and invaluable feedback. It brings us great joy to note that our work has captured the interest of the reviewers (R1 and R2), been recognized for its novelty (R1 and R3), and acknowledged as insightful (R1 and R2). Furthermore, we are delighted to learn that the matter of addressing knowledge consolidation in LLMs has been recognized as "an important topic that needs to be address" (R2). We have meticulously examined all the concerns raised by the reviewers and have diligently endeavored to address these issues, as detailed in the discussion below.
>
> **Reviewer’s Comment: “I wonder how the dataset and the task of knowledge consolidation relates to multi-hop question answering and datasets like HotpotQA. While Epik-Eval makes emphasis on a sequence of events with knowledge that needs to be consolidated in order to reach a final output, HotpotQA requires the model to extract information at each step of the reasoning chain that allows addressing such steps in order to answer the overall multi-hop question.”**
>
> **Response:** In the multi-hop question answering benchmarks, a model is given a question and can then search through multiple documents such that the model can use text from these documents  in order to answer the question. This is akin to providing entire documents in the prompt along with the question. Thus, the setup is pure inference and the model must consolidate external knowledge (information from the documents it is searching through). In contrast our work focuses on analyzing if these models can consolidate the knowledge stored in their parameter space (internal knowledge), that they have acquired through training. To do so, we train the models on documents, in order to memorize them, and then in inference mode, evaluate the models for their ability to consolidate and recall the necessary information from the multiple memorized documents in order to be able to answer the question. The focus is thus on the model's ability to consolidate the proper knowledge stored in the parameter space instead of evaluating its ability to search through documents. We will update the related work section to discuss these distinctions.
>
> **Reviewer’s Comment: “I would have liked to see more LLMs evaluated on the dataset in this paper”**
>
> **Response:** We have evaluated Llama-2 7b, which gets 49.56% accuracy when trained on unsegmented stories and 7.56% accuracy when trained on segmented stories. We are currently evaluating the other Llama-2 variants and will do the same with Llama 1 and the OPT models.

---

### Official Review · Reviewer_hoTF · 2023-08-05

**Soundness:** 3

**Excitement:**

4: Strong: This paper deepens the understanding of some phenomenon or lowers the barriers to an existing research direction.

**Missing References:**

[1] Liu, J., Shen, D., Zhang, Y., Dolan, B., Carin, L., & Chen, W. (2021). What Makes Good In-Context Examples for GPT-$3 $?. arXiv preprint arXiv:2101.06804.
[2]  Fu, Y., Peng, H., Sabharwal, A., Clark, P., & Khot, T. (2022). Complexity-based prompting for multi-step reasoning. arXiv preprint arXiv:2210.00720.
[3] Lu, Y., Bartolo, M., Moore, A., Riedel, S., & Stenetorp, P. (2021). Fantastically ordered prompts and where to find them: Overcoming few-shot prompt order sensitivity. arXiv preprint arXiv:2104.08786.

**Paper Topic And Main Contributions:**

The present paper proposes a dataset for evaluation of epistemic capabilities of language models by means of inspection of knowledge integration using several reasoning tasks of different. The work's main contribution lies in the proposal of the dataset, but also in the experimental analysis of the performance T5 and Flan-T5 models on the resource.
The authors propose a dataset composed of a collection of 1.8K narrative texts, transformed into 3.6K question-answer pair, for reasoning tasks of different nature probing the ability of models to consolidate information. The authors then employ this resource to evaluate the epistemic capabilities of T5 and Flan-T5 models of different size in two different training regimens. They strive for answering two guiding questions regarding the nature of knowledge representation and consolidation in these models the the effect of scaling on this capability. The authors further conduct analysis and discussion of their experimental results and pointing to how knowledge consolidation can be improved in language models.

**Questions For The Authors:**

How are the prompts for the segmented narratives constructed? In which sense do they differ from the non-segmented narratives?
What does Figure 3 mean? What is measured on X-axis in the graphs? Is is the size of the model (number of parameters)?
What do the authors mean by partial observability and full observability?

**Reasons To Accept:**

The proposed dataset is novel and can point to different avenues on how we can improve knowledge consolidation and encoding in LLM, especially for applications such as LM as KB and knowledge-rich tasks. Currently, complex structures have been employed such as graph NN and advanced neural machines to deal with the limitations of knowledge encoding and processing in neural models and information consolidation could indicate avenues to deal with such complex tasks in LLM in a more efficient manner.

TL;DR
Interesting new resource for probing capabilities of LLM;
To my knowledge, information consolidation capabilities in LLM have not been independently measured in any work, and this gives insights into the limitations of such models in LM as KB and reasoning settings in knowledge-rich scenarios.

**Reasons To Reject:**

While interesting in principle, several aspects of the work are not completely clear. Also, I have not been able to identify exactly the results achieved by the authors can be understood as probing of epistemic capabilities of language models in its interpretation of LLM as epistemic models. While the resource gives insights on the limitation of these models in reasoning tasks and of LLm both as knowledge bases and epistemic models, the presented evaluation tasks and analysis don't strike me as epistemic in nature - for example, it is false that the expected answer to question 2 in Table 1 should be no unless we understand epistemic reasoning in a rather limited setting. This difficulty may also lie in the fact that I have not been able to understand exactly how the in-context training with the segmented narrative texts works, as it was not particularly clear in the manuscript.
Overall, in terms of analysing the epistemic capabilities of LLM, I fail to see how that work informs us more than those regarding the neural theory of Mind, such as that of Sap et al. and Liu et al. Also, it is important to notice that CoT reasoning has been observed to be very susceptible to the examples presented [1,2] and the order of presentation of examples [3], but the failure to control for these and other important characteristics that could influence the performance of the models, yet they draw conclusions regarding the behaviour of such models. The authors also do not clearly discuss how and why each task was chosen and the methodological aspects of creation/selection and validation of examples in the corpus, and whether they are good proxies for information consolidation and epistemic reasoning. Without fully discussing the proposed resource, I hardly see how any of the authors' claims can be trusted. I do think the resource is interesting as a collection of different reasoning tasks that require information consolidation, and thus let us probe the consolidation powers of LLMs, but I find the authors' claims are not properly justified in the manuscript.
The work limits itself to the English language, but in no moment the authors consider how (syntactic, semantic, pragmatic, etc) aspects of the language can impact the performance of the models  - especially in a task for language-based reasoning.

TL;DR
The authors fail to clearly present the resource and methodological aspects for its creation and validation and discussion of the corpus composition in order for the reader to understand the empirical study resented later;
The conclusions drawn by the authors are, in my opinion, unsubstantiated by the work presented. The authors seem to draw much deeper conclusions than their experiments allow and seem to fail to control aspects that may explain the performance of the models in their experiments, but no do discuss these limitations in Limited to the English language.

**Reproducibility:**

4: Could mostly reproduce the results, but there may be some variation because of sample variance or minor variations in their interpretation of the protocol or method.

**Reviewer Confidence:**

3: Pretty sure, but there's a chance I missed something. Although I have a good feel for this area in general, I did not carefully check the paper's details, e.g., the math, experimental design, or novelty.

**Typos Grammar Style And Presentation Improvements:**

I would like to see better examples of how the training with segmented narratives works and some explanation of Figure 3, which I found very difficult to understand.

---

> ### Author Rebuttal · Authors · 2023-08-29
>
> We would like to extend our sincere appreciation to the reviewers for their thorough and invaluable feedback. It brings us great joy to note that our work has captured the interest of the reviewers (R1 and R2), been recognized for its novelty (R1 and R3), and acknowledged as insightful (R1 and R2). Furthermore, we are delighted to learn that the matter of addressing knowledge consolidation in LLMs has been recognized as "an important topic that needs to be address" (R2). We have meticulously examined all the concerns raised by the reviewers and have diligently endeavored to address these issues, as detailed in the discussion below.
>
> **Reviewer’s Comment: “...CoT reasoning has been observed to be very susceptible to the examples presented [1,2] and the order of presentation of examples [3], but the failure to control for these and other important characteristics that could influence the performance of the models, yet they draw conclusions regarding the behaviour of such models.”**
>
> **Response:** There seems to be a misunderstanding here, as we are not using few-shot examples or the chain-of-thought reasoning.
>
> **Reviewer’s Comment: “The authors also do not clearly discuss how and why each task was chosen and the methodological aspects of creation/selection and validation of examples in the corpus, and whether they are good proxies for information consolidation and epistemic reasoning.”**
>
> **Response:** When selecting the tasks, we tried to cover as many different types of relationship between facts and events as possible. By reading stories and writing down questions, we came up with a comprehensive list of 18 tasks. These general questions can then be rephrased in many different ways. Finally, we grouped them in categories, for example tasks about temporal relationships. These categories aren't perfect though, as tasks from different categories sometimes share similar aspects, such as counting. Thus the categories do contain some subjective aspects, but as for the tasks themselves, we believe they cover a broad range of questions about dependencies between facts and events in a story. Note that this choice is secondary to the objective of measuring the knowledge consolidation capabilities of these models, where the focus is on the consolidation and recall capabilities. We have added this explanation to the paper.
>
> **Reviewer’s Comment: “The work limits itself to the English language, but in no moment the authors consider how (syntactic, semantic, pragmatic, etc) aspects of the language can impact the performance of the models - especially in a task for language-based reasoning.”**
>
> **Response:** We agree that the nature of the language (syntax, semantics etc) could play a role in the performance of the model, as the performance of a machine learning system does depend on the properties of the dataset it is trained on. However, in this case, the effect (if any) should be quite limited as we are finetuning the models on datasets from the same language. In that sense, the choice of the language is secondary.
>
> **Reviewer’s Comment: “... seem to fail to control aspects that may explain the performance of the models in their experiments.”**
>
> **Response:** We trained the models on our dataset in the same way as described in the papers they were introduced in, i.e. the models see the same stories multiple times during training in random order. We decided to replicate the pretraining methodology as much as possible, since the goal is to make some parallels about how LMs consolidate knowledge from various documents during training. This is so we do not introduce a distributional shift, which could become a confounder. At test time, there are no few-shot examples and no chain-of-thought reasoning is applied. This is a reasonable choice as the main objective is to study if these models are capable of piecing back the story in the proper order and not to evaluate the "difficulty" of the reasoning aspect of the benchmark..
>
> **Reviewer’s Comment: "I find the authors' claims are not properly justified in the manuscript." and "The authors seem to draw much deeper conclusions than their experiments allow…"**
>
> **Response:** We are confused as to which claims the reviewer is talking about, as our paper’s main claim is that LM’s struggle to consolidate their knowledge. Our results back up this claim and the reviewer themselves agree with us that we provide insight into the knowledge consolidation capabilities of LMs: “To my knowledge, information consolidation capabilities in LLM have not been independently measured in any work, and this gives insights into the limitations of such models in LM as KB and reasoning settings in knowledge-rich scenarios.” Additionally, no other reviewer has mentioned any issues with unjustified claims. Could the reviewer elaborate on which claims exactly they felt were not well justified?
>
> **Reviewer’s Comment: “How are the prompts for the segmented narratives constructed? In which sense do they differ from the non-segmented narratives?”**
>
> **Response:** There are two types of prompts: stories to memorize and question-answers. The format of stories is as follows: For unsegmented stories:
> > <A title/header>
> >
> > <The story>
>
> So for example you could have:
> > [Task 1] Tom's Afternoon
> >
> > 1 PM: Tom reads a book.
> >
> > 3 PM: Tom goes to the store.
> >
> > 5 PM: Tom is at the restaurant.
>
> The segmented stories follow the same format, so for this example, we would have the three following text sequences instead of only one:
> > [Task 1] Tom's Afternoon, Part 1/3
> >
> > 1 PM: Tom reads a book.
>
> > [Task 1] Tom's Afternoon, Part 2/3
> >
> > 3 PM: Tom goes to the store.
>
> > [Task 1] Tom's Afternoon, Part 3/3
> >
> > 5 PM: Tom is at the restaurant.
>
> As for the format of question-answer prompts, they follow this general format:
> > <Question>
> >
> > <Recall the story> <Reasoning part for certain tasks> <Final answer>
>
> For example:
> > [Task 1] Did Tom go to the library during his afternoon?
> >
> > 1 PM: Tom reads a book. 3 PM: Tom goes to the store. 5 PM: Tom is at the restaurant. The answer is no.
>
> As we explain in the paper, one version of the model is trained on the unsegmented version of the stories while the other version is trained on the segmented version. Both versions are also trained on question-answer examples, which refer to stories not included in the training set. Their purpose is to help the models learn the appropriate answer format.
>
> **Reviewer’s Comment: “What does Figure 3 mean? What is measured on X-axis in the graphs? Is it the size of the model (number of parameters)?”**
>
> **Response:** Figure 3 simply breaks down the accuracy of the models per category of tasks. In Figure 2, the shown accuracy is the global accuracy. In Figure 3 we simply break it down per category. We also only show the best and worst performance of models trained on unsegmented stories to simplify the plots. The x-axis is the number of parameters of these models, so it captures the size of the models.
>
> **Reviewer’s Comment: “What do the authors mean by partial observability and full observability?”**
>
> **Response:** The inspiration to use these terms comes from world modeling and reinforcement learning. In our work, they are equivalent terms for the segmented setup (partial observability) and the unsegmented setup (full observability), meaning that in one case models are trained on segmented narratives and in the other, models are trained on unsegmented narratives. To avoid any confusion in the paper though, we will replace this terminology with the unsegmented/segmented setup.
>
> **Reviewer’s Comment: “I have not been able to identify exactly the results achieved by the authors can be understood as probing of epistemic capabilities of language models in its interpretation of LLM as epistemic models.”**
>
> **Response:** There seems to be a misunderstanding here. When we claim that we want to study language models as epistemic models, we mean studying their knowledge consolidation capabilities, not the formal definition of epistemology. We have updated section 2 in order to clarify this, see below.
>
> ## 2 Epistemology & Language Models
> Epistemic frameworks (Wang, 2015; Rendsvig and Symons, 2019) are formal systems used to represent knowledge, belief and the uncertainty that entails what a reasoning system knows and/or believes. This is enabled through organizing the knowledge observed by the system. The rules to combine the knowledge in the abstract framework governs combining a new information to the current set of information, or when to ignore the new information, and using the current beliefs to anticipate related events. While LMs behave as KBs to store known relations, epistemic logic provides us with the inspiration to describe how these models organize and update their knowledge.
>
> Consider the example from Figure 1, where we have the knowledge x_1: “Tom ate an apple.”, x_2: “Tom ate a pear.” and x_3: “Bob ate an orange.”. Prompted with the question “What did Tom eat?”, the model must recall knowledge from within its parameter space. It has to connect x_1 and x_2 while also ignoring x_3. To answer the query, a system is expected to consolidate the information and retain a knowledge state over the information it had seen until then. However, an inability to draw the connections would leave the facts disconnected. We describe the model that struggles to consolidate as Type I, and one that is better at it and infer over a consolidated knowledge state as Type II.
>
> As LMs are used in many real-world scenarios that are frequently presented with a periodic flow of information, it is necessary for the LMs to use such information appropriately during inference. While techniques like self-prompting and generation over self-retrieval are gaining popularity, the performance relies on the quality of the prompt, which adds to the robustness concerns on the performance of LMs on varying reasoning tasks. Inspired by epistemology, we design Epik-Evalto to diagnose whether LMs comply with a first-order knowledge state following a sequence of facts which holds a consolidated summary of information during inference.
>
> **Reviewer’s Comment: “I have not been able to understand exactly how the in-context training with the segmented narrative texts works, as it was not particularly clear in the manuscript.”**
>
> **Response:** We have rewritten section 3 in order to clarify the description of the dataset, the details on its generation, and how the models are evaluated in detail. See the updated methodology section 3 that follows.
>
> ## 3 EpiK-Eval
> The EpiK-Eval benchmark presents a suite of novel, narrative-based diagnostic tasks, meticulously designed to evaluate a LM's capacity to construct a comprehensive, unified knowledge state.
>
> **Dataset:** Our benchmark comprises 18 tasks, which are questions about relations between facts and events in stories, for example: "Does x happen before/after y?". See Table 2 for the full list of tasks. For each of these, we generate 100 stories following a per task template. For example, Task 2 has the following template:
> > [Task 2] {name}'s Vacation
> >
> > {name} went {activity} on {day}.
> >
> > .
> >
> > .
> >
> > .
>
> where the first line is the story's title, the name is randomly sampled such that it is unique to each story and the activity and day in a sentence are randomly sampled from the list ['fishing', 'hiking'] and ['Monday', 'Tuesday', 'Wednesday', 'Thursday', 'Friday', 'Saturday', 'Sunday'] respectively. The story can have a random number of sentences, the range is pre-determined per task, for example, Task 2 stories can have between 3 and 5 sentences. Here is an example of a generated story for Task 2:
> > [Task 2] Tom's Vacation
> >
> > Tom went fishing on Monday.
> >
> > Tom went hiking on Wednesday.
> >
> > Tom went fishing on Saturday.
>
> Thus, with a 100 stories generated for each 18 tasks, we get 1800 stories, which we refer to as our dataset of unsegmented stories D_U = {x_1, x_2, ..., x_1800}. After having generated these stories, we also generate a second dataset which consists of the segmented version of these stories. For each given story, we segment it into its individual sentences and add a part number to the title. For example, given the previous story about Tom, we would get the following three text sequences:
> > [Task 2] Tom's Vacation, Part 1/3
> >
> > Tom went fishing on Monday.
>
> > [Task 2] Tom's Vacation, Part 2/3
> >
> > Tom went hiking on Wednesday.
>
> > [Task 2] Tom's Vacation, Part 3/3
> >
> > Tom went fishing on Saturday.
>
> We do this for all 1800 stories and get 6800 story segments, which form our dataset of story segments D_S = {s_1, s_2, ..., s_6800}.
>
> For each story, we also generate one question-answer pair. The questions are rephrasings of the task. For example, for Task 2 "How many times does x happen?", we have "How many times did {name} go fishing?". The question-answer pairs are also generated following a template. The template always consists of a question followed by the answer which itself has three parts: recall of the entire story, an optional reasoning part depending on the task and the final answer. For example, the template for question-answers in Task 2 is the following:
> > [Task 2] How many times did {name} go fishing?
> >
> > {story} The answer is {count}.
>
> and here is an example of a generated question-answer pair:
> > [Task 2] How many times did Tom go fishing?
> >
> > Tom went fishing on Monday. Tom went hiking on Wednesday. Tom went fishing on Saturday. The answer is 2.
>
> We also include in the Appendix a full list of templates as well as the values each placeholder variable (e.g., activities, days) can take, in order to generate stories and question-answer pairs for each task. Having generated one question per story, we have a total of 1800 question-answer pairs, which we randomly split into two sets: the validation and the test set. In order for the models to learn the answer format that they are expected to respect while answering the questions, we add question-answer examples to the training set. We thus generate an additional 1800 stories and question-answer pairs. We discard the stories and add these 1800 question-answer pairs to the training set, such that there are no overlaps between questions in the training, validation and test set.
>
> **Evaluation Process:** We evaluate pretrained LMs for their ability to consolidate knowledge. To do so, given a pretrained language model, we make two copies of it: M_U and M_S. We fine-tune M_U on the unsegmented stories and M_S on the segmented stories. The prior setting allows for all the information necessary to answer a given question to be found in a single text sequence without requiring the model to learn dependencies across multiple text sequences, while the latter setting requires consolidating the information from the narrative segments. Having both allows to measure the effect of information being spread across separate text sequences and the LMs' ability to consolidate this knowledge during inference, by measuring the gap in performance between both models.
>
> Both models are fine-tuned on their respective dataset, D_U and D_S, as well as the training set of question-answer examples. Thus, one epoch for M_U consists of 3600 samples (1800 stories + 1800 q/a examples) and one epoch for M_S of 8600 samples (6800 segments + 1800 q/a examples). Samples are shuffled such that a batch may contain a mix of stories and question-answer examples in the case of M_U or story segments and question-answer examples in the case of M_S. Models are fine-tuned with their respective pretraining objective. Specifically, in the case of encoder-decoder style models, the story's title (first line in the text sequence) is fed to the encoder and the decoder is expected to output the rest of the story in the case of M_U or the story segment in the case of M_S. As for question-answer pairs, the question is fed to the encoder and the model is expected to output the answer. For causal language models, they are simply expected to predict the next token in the given sequence, as is standard procedure. Precisely, for M_U, a text sequence is either an entire story or a question concatenated with its answer, while for M_S, a text sequence is either a story segment or a question concatenated with its answer.
>
> During fine-tuning, both models are also periodically evaluated on the validation set. Models are run in inference mode as described in the papers they were introduced in. We prompt the models with the questions from the validation set and their answers are compared to the target answers. For an answer to be deemed as correct, it must match the exact target answer. This is to capture potential recall and reasoning errors as well as verifying the final answer. It is important for us to evaluate M_S's ability to consolidate the separate story segments, which is why we require the model to recall the entire story when answering a question. Here, M_U serves as an upper-bound on the performance and any potential gap in performance between it and M_S showcases the added difficulty of consolidating knowledge from the story segments. The number of correctly answered questions over the total number of questions is referred to as the accuracy. We also measure an additional metric, which we refer to as the hallucination rate. Given an answer, consider only the recall part of the answer and disregard the reasoning part and the final answer. Our hallucination rate is the number of recalled sentences that contains an error (does not match with the actual sentence in the narrative) over the total number of recalled sentences. This metric provides a more fine-grained examination of the recall and knowledge consolidation capabilities of the model. We want to evaluate if the model is more likely to hallucinate a fact, event or segments when recalling these from multiple training sequences (segmented setup) versus a single training sequence (unsegmented setup).
>
> Once both models have been fine-tuned, we take the best performing checkpoint of each model on the validation set and evaluate these on the test set. This is done in the same manner as the validation, except that the questions are from the test set.

---

### Meta-Review · Area_Chair_2Ztm · 2023-09-10

**Recommendation:** 5

**Metareview:**

The main conclusions of the reviews and the post-rebuttal discussions:
- 3/ 3 reviewers consider the paper sound (scores 3, 3, 3)
- 3/ 3 reviewers find the paper exciting (scores 4, 4, 3)

From reading the rebuttal and seeing the scores above, I find that the reviewers consider strong points for soundness the following:
- Novel dataset for evaluation of epistemic capabilities of language models using inspection of knowledge integration using several reasoning tasks.
- Good experimental analysis of the performance T5 and Flan-T5 models on the dataset and final recommendations
- Knowledge consolidation in LLMs is important. The ability to combine information seen in different documents during training is crucial for many applications. This is the first study that investigates LMs’ capability to perform this information combination effectively.

---

### Decision · Program_Chairs · 2023-10-07

**Decision:**

Accept-Main

**Comment:**

The main conclusions of the reviews and the post-rebuttal discussions:
- 3/ 3 reviewers consider the paper sound (scores 3, 3, 3)
- 3/ 3 reviewers find the paper exciting (scores 4, 4, 3)

From reading the rebuttal and seeing the scores above, I find that the reviewers consider strong points for soundness the following:
- Novel dataset for evaluation of epistemic capabilities of language models using inspection of knowledge integration using several reasoning tasks.
- Good experimental analysis of the performance T5 and Flan-T5 models on the dataset and final recommendations
- Knowledge consolidation in LLMs is important. The ability to combine information seen in different documents during training is crucial for many applications. This is the first study that investigates LMs’ capability to perform this information combination effectively.